

# A dual-pass carbon cycle data assimilation system to estimate surface $CO_2$ fluxes and 3D atmospheric $CO_2$ concentrations from spaceborne measurements of atmospheric $CO_2$

Rui Han[1,2], Xiangjun Tian[1,2,3]

[1]International Center for Climate and Environment Sciences, Institute of Atmospheric Physics, Chinese Academy of Sciences, Beijing, 100029, China
[2]University of Chinese Academy of Sciences, Beijing, 100049, China
[3]Collaborative Innovation Center on Forecast and Evaluation of Meteorological Disasters, Nanjing University of Information
Science and Technology, Nanjing, 210044, China

*Correspondence to*: Xiangjun Tian (tianxj@mail.iap.ac.cn)

**Abstract.** Here we introduce a new version of the carbon cycle data assimilation system, Tan-Tracker (v1), which is based on the Nonlinear Least Squares Four-dimensional Variational Data Assimilation algorithm (NLS-4DVar) and the Goddard Earth Observing System atmospheric chemistry transport model (GEOS-Chem). Using a dual-pass assimilation framework

that consists of a carbon dioxide ($CO_2$) assimilation pass and a flux assimilation pass, we assimilated the atmosphere column-averaged $CO_2$ dry air mole fraction ($XCO_2$), while sequentially optimizing the $CO_2$ concentration and surface carbon flux via different length windows with the same initial time. When the $CO_2$ assimilation pass is first performed, a shorter window of 3 days is applied to reduce the influence of the background flux on the initial $CO_2$ concentration. This allows us to obtain a better initial $CO_2$ concentration to drive subsequent flux assimilation passes. In the following flux assimilation

pass, a properly elongated window of 2 weeks absorbs enough observations to reduce the influence of the initial $CO_2$ concentration deviation on the flux, resulting in better surface fluxes. In contrast, the joint assimilation system Tan-Tracker (v0) uses the same assimilation window for optimization of $CO_2$ concentration and flux, making the uncertainties in $CO_2$ concentration and flux indistinguishable. The proper orthogonal decomposition (POD)-4DVar algorithm applied with the older system is only a rough approximation of the one-step iteration of the NLS-4DVar algorithm; thus, it can be difficult to

fully resolve the nonlinear relationship between flux and $CO_2$ concentration. In this study, we designed a set of observation system simulation experiments to assimilate artificial $XCO_2$ observations, in an attempt to verify the performance of the newly developed dual-pass Tan-Tracker (v1). Compared with the prior and joint system, the dual-pass system provided a better representation of the spatiotemporal distribution of the true flux and true $CO_2$ concentration. We performed sensitivity tests of the flux assimilation window length and number of NLS-4DVar assimilation iterations. Our results indicated that the

appropriate flux assimilation window length (14 days) and the appropriate number of NLS-4DVar maximum iterations (three) could be used to achieve optimal results. Thus, the Tan-Tracker (v1) system, based on a novel dual-pass assimilation framework, provides more accurate surface flux inversion estimates and is ultimately a better tool for carbon cycle research.



## 1. Introduction

Since the Industrial Revolution, humans have consumed fossil fuels and emitted large amounts of carbon dioxide ($CO_2$). About 50% of the $CO_2$ remains in the atmosphere. The continuous rise in global atmospheric $CO_2$ concentrations breaks the radiation balance of the Earth system, resulting in global climate change. The remaining $CO_2$ is absorbed by the terrestrial

ecosystem and oceans; however, there are still many uncertainties associated with these absorption mechanisms (Ballantyne et al., 2012; Le Quéré et al., 2017). Determining the appropriate carbon budget for the Earth's ecosystem and oceans is important for the development of relevant climate policies and predictions of future scenarios, having been the focus of extensive carbon cycle research (Stocker et al., 2013). In recent years, there has been an increase in multi-source atmospheric $CO_2$ concentration measurements and model development. The surface carbon flux inversion method, obtained by

combining model and atmospheric $CO_2$ information, has made great progress in carbon cycle data assimilation (Peters et al., 2005; Peters et al., 2007; Tian et al., 2014; Deng et al., 2016; Feng et al., 2016; Basu et al., 2013;  Basu et al., 2018).

Many have attempted to optimize surface carbon flux measurements. For example, Carbon-Tracker (Peters et al., 2005; Peters et al., 2007) is a well-designed carbon assimilation system that uses Transport Model 5 (TM5) and the ensemble Kalman filter (EnKF) method (Evensen, 1994) to assimilate *in situ* $CO_2$ observations. The Carbon Cycle Data Assimilation

System (CCDAS) (Rayner et al., 2005) Kaminski et al., 2013) couples the Biosphere Energy-Transfer HYdrosphere (BETHY) model (Kaminski and Heimann, 2001) with the atmospheric transport model TM2, to assimilate satellite observations of photosynthetically active radiation and atmospheric $CO_2$ concentration observations; the approach is a two-step process, in which the parameters of the carbon cycle model are first optimized to improve surface flux measurement accuracy. Tan-Tracker (v0) (Tian et al., 2014) uses the Goddard Earth Observing System atmospheric chemistry transport

model (GEOS-Chem) and the identity matrix as a joint dynamical model; a proper orthogonal decomposition (POD)-based four-dimensional variational assimilation algorithm (POD-4DVar) (Tian et al., 2011) is combined with a joint assimilation framework to integrate *in situ* $CO_2$ concentration observations, with simultaneous optimization of the $CO_2$ concentration and flux. This method has obtained good results; however, there are still some problems associated with the joint assimilation framework. The same window lengths limit the ability to distinguish the $CO_2$ concentration from the flux. Additionally, the

POD-4DVar algorithm is only a rough approximation of a one-step iteration of the Nonlinear Least Squares (NLS)-4DVar algorithm (Tian and Feng, 2015; Tian et al., 2018). Although the above assimilation system has achieved reasonable results, the sparse and uneven spatial distributions of *in situ* stations greatly limit the flux optimization accuracy. Several unconventional data assimilation techniques have been explored. For example, Zhang et al. (2014) conducted an assimilation of the aircraft observation Comprehensive Observation Network for Trace gases by Airline (CONTRAL) based on Carbon-

Tracker.

With the launch of the Greenhouse gases Observing SATellite (GOSAT) (Kuze et al., 2009) and the Orbiting Carbon Observatory-2 (OCO-2) satellite (Crisp et al., 2017), satellite data assimilation experiments have also been conducted based on the atmosphere column-averaged $CO_2$ dry air mole fraction ($XCO_2$) at higher temporal and spatial resolutions. Basu et al.



(2013) used TM5 4DVar to assimilate GOSAT observations, and showed that satellite data provided an effective constraint for surface carbon source–sink inversion. Tian et al. (2014) used Tan-Tracker (v0) to conduct GOSAT observation assimilations using a set of observing system simulation experiments (OSSEs), and found that the optimized $CO_2$ concentration and flux showed expected results. Deng et al. (2016) used the GEOS-Chem and the 4DVar method to

simultaneously assimilate GOSAT observations of the land and ocean. This method provided a better representation of the $CO_2$ surface flux than others that used only terrestrial observations; additionally, the results indicated that increasing the observation coverage further improved the sensitivity of surface flux inversion measurements. Feng et al. (2016) used the EnKF to assimilate GOSAT observations in Europe; the flux inversion results obtained displayed a larger amplitude change than those using an *in situ* station. Basu et al. (2018) applied 4DVar OSSEs to OCO-2 observations with multiple

atmospheric transport models; they showed that the wider global coverage provided by OCO-2 observations enabled better surface flux representation than *in situ* observations. Overall, flux results depend on the atmospheric chemical transmission mode used. The abovementioned assimilation attempts using satellite data have reduced the uncertainty associated with flux measurements and provided some insight into surface carbon flux mechanisms. However, the assimilation of satellite column-average concentration observations of $XCO_2$ is still in the exploratory stage.

Based on GEOS-Chem and NLS-4DVar (Tian et al., 2018) assimilation of $XCO_2$ satellite observations, we introduce the Tan-Tracker (v1) carbon cycle data assimilation system. The novel dual-pass data assimilation framework consists of a $CO_2$ assimilation pass and a flux assimilation pass, which have the same initial time but different assimilation window lengths. Specifically, the first $CO_2$ assimilation pass uses a shorter window of 3 days to reduce the influence of background flux on the initial $CO_2$ readings. By minimizing the initial $CO_2$ deviation, better initial $CO_2$ concentrations are derived for the

subsequent flux assimilation pass. In the following flux assimilation pass, a properly elongated window of 2 weeks absorbs enough observations to reduce the influence of the initial $CO_2$ concentration deviation on the flux, resulting in a better representation of the surface flux. Compared with the joint Tan-Tracker (v0) assimilation system, the Tan-Tracker (v1) system uses a dual-pass framework to mitigate the effects of the initial $CO_2$ concentration on surface flux, while using a more advanced assimilation algorithm, NLS-4DVar, to improve the accuracy of the optimized flux results.

This paper is divided into four sections. Section 2 introduces the method and the framework of the Tan-Tracker (v1) system and its coupling to the NLS-4DVar algorithm. In Section 3, we describe the OSSE design using OCO-2 observations, and compare Tan-Tracker (v1), Tan-Tracker (v0), and control experimental results to true results. The flux obtained using Tan-Tracker (v1) exhibited a total spatiotemporal flux distribution and optimized $CO_2$ concentration that were closer to those of the true flux. A summary and conclusions are presented in Section 4.



## 2. Methods and Systems

### 2.1 Dual-pass Tan-Tracker (v1) assimilation system framework

The dual-pass carbon cycle data assimilation system Tan-Tracker (v1) is divided into two assimilation passes: a $CO_2$ assimilation pass and a flux assimilation pass, in addition to an update section (Fig. 1). Based on the NLS-4DVar (Tian and

Feng, 2015; Tian et al., 2018) assimilation method for satellite column-average $CO_2$ concentration measurements of $XCO_2$, we optimized the $CO_2$ concentration and surface $CO_2$ flux in different lengths of assimilation windows with the same initial time $t_0$ of $CO_2$ concentration. First, the $CO_2$ assimilation pass is implemented. The shorter 3-day window reduces the influence of background flux on the initial $CO_2$ measurements, minimizing the initial $CO_2$ deviation to obtain a better initial $CO_2$ concentration to drive the flux assimilation pass. In the following flux assimilation pass, a properly elongated window

of 2 weeks absorbs enough observations to reduce the influence of the initial $CO_2$ concentration deviation on the flux. The evolution of the $CO_2$ concentration in the assimilation window is dominated by the flux for improved accuracy of surface flux measurements. The update section guarantees a connection between the two adjacent assimilation windows, in which the initial $CO_2$ concentration and background flux of the $CO_2$ assimilation pass are provided for the next window, allowing the background flux and flux ensembles of the flux assimilation pass to be updated.

The $CO_2$ assimilation pass is shown in the blue portion of Figure 1. Given that NLS-4DVar is an ensemble-based hybrid assimilation algorithm, we first prepared a set of 3-day-length $CO_2$ concentration ensembles, $\mathbf{U}_{s,i}, (i = 1, \cdots, N)$ (see Section 2.3), where $S$ denotes the ensembles and $N$ is the ensemble number. In the $CO_2$ assimilation pass, we used $N = 160$. Starting from the background initial $CO_2$ $\mathbf{U}_{b,t_0}$ forcing by the background flux:

$$\mathbf{F}_b = \boldsymbol{\lambda}_b \times \mathbf{F}^*, \tag{1}$$

where $\mathbf{F}^*$ is the prior flux and $\boldsymbol{\lambda}_b$ is a linear scale factor (Peters et al., 2005; Tian et al., 2014) for the assimilation window, we simulated the 3-day $CO_2$ concentration $\mathbf{U}_b$ used as the background $CO_2$. $H_k$ is a satellite $XCO_2$ observation operator, as given in Eq. (31). Putting $\mathbf{U}_s, \mathbf{U}_b, H_k$ together with observations $X_{CO_2,Obs}$ into the NLS-4DVar processor, we can obtain an optimized initial $CO_2$ $\mathbf{U}_{a,t_0}$, to be used as the initial $CO_2$ of the flux assimilation pass.

In the flux assimilation pass (the red portion shown in Fig. 1), we assume that there is no error in anthropogenic emissions,

and only optimize the terrestrial ecosystems flux and oceans flux:

$$\mathbf{F}^* = \mathbf{F}^*_{bio} + \mathbf{F}^*_{oce}, \tag{2}$$

where $\mathbf{F}^*$ is the prior flux, with $bio$ referring to the flux from the terrestrial biosphere, and $oce$ representing the flux from the ocean. Starting from the optimized initial $CO_2$ $\mathbf{U}_{a,t_0}$, forcing by a set of prepared flux ensembles:

$$\mathbf{F}_{s,i} = \boldsymbol{\lambda}_{s,i} \times \mathbf{F}^*, (i = 1, \cdots, N), \tag{3}$$



we obtain a set of 2-week $CO_2$ ensembles $\mathbf{U}_{s,i}, (i = 1, \cdots, N)$, where $\boldsymbol{\lambda}_{s,i} (i = 1, \cdots, N)$ is a set of scale factors (see Section 2.3). Using an optimization variable for the flux and considering computational cost, we chose $N = 36$. Simultaneously, starting from the background initial $CO_2$ $\mathbf{U}_{b,t_0}$ forcing by the background flux $\mathbf{F}_b = \boldsymbol{\lambda}_b \times \mathbf{F}^*$, we simulated the 2-week $CO_2$ concentration $\mathbf{U}_b$ as background $CO_2$. Putting $\boldsymbol{\lambda}_s, \mathbf{U}_s, \boldsymbol{\lambda}_b, \mathbf{U}_b, H_k$, and observations $X_{CO_2, Obs}$ into the NLS-4DVar

processor, we can obtain the optimized scale factor $\boldsymbol{\lambda}_a$, with the optimized flux given by $\mathbf{F}_a = \boldsymbol{\lambda}_a \times \mathbf{F}^*$.

The update section is shown as the black portion of Figure 1. Starting from the optimized initial $CO_2$ $\mathbf{U}_{a,t_0,r}$ of the $r$th assimilation cycle forcing by optimized fluxes $\mathbf{F}_{a,r}$, and integrating through the window of the flux assimilation pass to the end, we obtain the background initial $CO_2$ concentration $\mathbf{U}_{b.t_0,r+1}$ of the $(r+1)$th assimilation cycle. Unlike the joint Tan-Tracker (v0) system, the background initial $CO_2$ concentration of Tan-Tracker (v1) is obtained by a running model, as

opposed to a direct assimilation, thus eliminating the problem of $CO_2$ over-optimization. Similar to the approach of Peters (2007), the $(r+1)$th background flux, $\mathbf{F}_{b,r+1} = \boldsymbol{\lambda}_{b,r+1} \times \mathbf{F}^*_{r+1}$, is applied using the mean value of the two previous time steps' scale factors $a$:

$$\boldsymbol{\lambda}_{b,r+1} = (\boldsymbol{\lambda}_{a,r} + \boldsymbol{\lambda}_{a,r-1} + 1) / 3. \tag{4}$$

## 2.2 Coupling of NLS-4DVar with Tan-Tracker (v1) assimilation framework

The NLS-4DVar algorithm is used to solve the optimal initial perturbation $\mathbf{x}^{'}_a$ to satisfy the incremental form of the 4DVar cost function:

$$J\left(\mathbf{x}^{'}\right) = \frac{1}{2}\left(\mathbf{x}^{'}\right)^{\mathbf{T}} \mathbf{B}^{-1}\left(\mathbf{x}^{'}\right) + \frac{1}{2}\sum_{k=0}^{S}\left[L^{'}_k\left(\mathbf{x}^{'}\right) - \mathbf{y}^{'}_{obs,k}\right]^{\mathbf{T}} \mathbf{R}^{-1}_k \left[L^{'}_k\left(\mathbf{x}^{'}\right) - \mathbf{y}^{'}_{obs,k}\right], \tag{5}$$

where $\mathbf{x}^{'} = \mathbf{x} - \mathbf{x}_b$ is the perturbation of the background field $\mathbf{x}_b$ at initial time $t_0$, and

$$L^{'}_k\left(\mathbf{x}^{'}\right) = L_k\left(\mathbf{x}_b + \mathbf{x}^{'}\right) - L_k\left(\mathbf{x}_b\right), \tag{6}$$

$$\mathbf{y}^{'}_{obs,k} = \mathbf{y}_{obs,k} - L_k\left(\mathbf{x}_b\right), \tag{7}$$

$$L_k = H_k M_{t_0 \to t_k}, \tag{8}$$

where the superscript $\mathbf{T}$ is the matrix transpose, the subscript $b$ is the background value, $\mathbf{y}_{obs,k}$ is the observation at time $t_k, k = 0,1,\cdots,S$, $H_k$ is the observation operator, $M_{t_0 \to t_k}$ is the nonlinear forecast model integrating from $t_0$ to $t_k$, and $\mathbf{B}$ and $\mathbf{R}_k$ are the background error and observational error covariance matrices, respectively. For simplicity,

$\mathbf{R} = diag(\mathbf{R}_0, \mathbf{R}_1, \cdots, \mathbf{R}_S)$.



As an ensemble-based assimilation approach, NLS-4DVar (Tian and Feng, 2015; Tian et al., 2018) assumes that the optimal analysis increment $\mathbf{x}_a^{'}$ can be expressed by a linear combination of the pre-prepared initial perturbations (IPs):

$$\mathbf{x}_a^{'} = \mathbf{P}_x \boldsymbol{\beta}, \tag{9}$$

where $\mathbf{P}_x = \left( \mathbf{x}_1^{'}, \mathbf{x}_2^{'}, \cdots, \mathbf{x}_N^{'} \right)$ are the initial perturbations, $\mathbf{x}_i^{'} = \mathbf{x}_i - \mathbf{x}_b, (i = 1, 2, \cdots, N)$, $N$ is the ensemble number, and

$\boldsymbol{\beta} = \left( \boldsymbol{\beta}_1, \boldsymbol{\beta}_2, \cdots, \boldsymbol{\beta}_N \right)$. We can replace the background error covariance matrix $\mathbf{B}$ with an ensemble perturbation estimate:

$$\mathbf{B}_e = \frac{\mathbf{P}_x \mathbf{P}_x^{\mathbf{T}}}{N-1}, \tag{10}$$

Furthermore, symmetric $\mathbf{R}$ has the Cholesky factorization,

$$\mathbf{R} = \mathbf{R}_+^{1/2} \left( \mathbf{R}_+^{1/2} \right)^{\mathbf{T}}, \tag{11}$$

Substituting Eqs. (9), (10) and (11) into Eq. (5), it can be rewritten as follows (Dennis and Schnabel, 1996),

$$J(\boldsymbol{\beta}) = \frac{1}{2} Q(\boldsymbol{\beta})^{\mathbf{T}} Q(\boldsymbol{\beta}), \tag{12}$$

$$Q(\boldsymbol{\beta}) = \begin{pmatrix} \mathbf{R}_+^{1/2} \left[ L^{'} \left( \mathbf{P}_x \boldsymbol{\beta} \right) - \mathbf{y}_{obs}^{'} \right] \\ \sqrt{N-1} \boldsymbol{\beta} \end{pmatrix}. \tag{13}$$

Thinking approximations (Tian and Feng, 2015):

$$L^{'} \left( \mathbf{x}_j^{'} \right) = \mathbf{y}_j^{'} \approx \mathbf{H}^{'} \mathbf{M}^{'} \left( \mathbf{x}_j^{'} \right), \tag{14}$$

and

$$L^{'} \left( \mathbf{P}_x \boldsymbol{\beta} \right) \approx \mathbf{H}^{'} \mathbf{M}^{'} \left( \mathbf{P}_x \boldsymbol{\beta} \right) \approx \mathbf{P}_y \boldsymbol{\beta}, \tag{15}$$

the first-derivate matrix (or Jacobian matrix) $J_{ac} Q(\boldsymbol{\beta})$ of $Q(\boldsymbol{\beta})$ can be computed approximately as follows,

$$J_{ac} Q(\boldsymbol{\beta}) = \frac{\partial Q(\boldsymbol{\beta})}{\partial \boldsymbol{\beta}} \approx \begin{pmatrix} \mathbf{R}_+^{1/2} \mathbf{P}_y \\ \sqrt{N-1} \mathbf{I} \end{pmatrix}, \tag{16}$$

where $\mathbf{I}$ denotes the $N \times N$ identity matrix. The Gauss-Newton iteration for the non-linear least squares problem (12) is defined by (Dennis and Schnabel, 1996):

$$\boldsymbol{\beta}^i = \boldsymbol{\beta}^{i-1} - \left[ \left( J_{ac} Q(\boldsymbol{\beta}^{i-1}) \right)^{\mathbf{T}} \left( J_{ac} Q(\boldsymbol{\beta}^{i-1}) \right) \right]^{-1} \left( J_{ac} Q(\boldsymbol{\beta}^{i-1}) \right)^{\mathbf{T}} Q(\boldsymbol{\beta}^{i-1}), \tag{17}$$

Substituting Eqs. (13) and (16) into Eq. (17), the cost function Eq. (5) can be rewritten as the least squares form of the control variable $\boldsymbol{\beta}$ (Tian and Feng, 2015) :

$$\boldsymbol{\beta}^i = \boldsymbol{\beta}^{i-1} + \left( \mathbf{P}_y^* \right)^{\mathbf{T}} L^{'} \left( \mathbf{x}_a^{'i-1} \right) + \left( \mathbf{P}_y^{\#} \right)^{\mathbf{T}} \mathbf{R}^{-1} \left[ \mathbf{y}_{obs}^{'} - L^{'} \left( \mathbf{x}_a^{'i-1} \right) \right], \tag{18}$$

$$\left( \mathbf{P}_y^* \right)^{\mathbf{T}} = -(N-1) \left[ \mathbf{P}_y^{\mathbf{T}} \mathbf{R}^{-1} \mathbf{P}_y + (N-1) \mathbf{I} \right]^{-1} \left[ \mathbf{P}_y^{\mathbf{T}} \mathbf{P}_y \right]^{-1} \mathbf{P}_y^{\mathbf{T}}, \tag{19}$$





$$\left(\mathbf{P}_y^{\#}\right)^{\mathbf{T}} = \left[\mathbf{P}_y^{\mathbf{T}}\mathbf{R}^{-1}\mathbf{P}_y + (N-1)\mathbf{I}\right]^{-1}\mathbf{P}_y^{\mathbf{T}}. \tag{20}$$

Here, $i = 1, 2, \cdots, I_{max}$, where $I_{max}$ is the maximum NLS-4DVar iteration number, $\mathbf{P}_y = (\mathbf{y}_1', \mathbf{y}_2', \cdots, \mathbf{y}_N')$, are the observation perturbations (OPs), $\mathbf{y}_j' = L'(\mathbf{x}_j'), (j = 1, 2, \cdots, N)$, and $L' = (L_0'^{\mathbf{T}}, L_1'^{\mathbf{T}}, \cdots, L_S'^{\mathbf{T}})$.

Using an ensemble-estimated $\mathbf{B}_e$ to replace the background error covariance matrix $\mathbf{B}$ will bring a spurious correlation that

can be eliminated by a localization scheme. An efficient local correlation matrix decomposition approach (Zhang and Tian, 2018) can be used to quickly assimilate a large number of observations while ensuring the assimilation results, especially for satellite data assimilation with high spatiotemporal resolution. Its implementation in NLS-4DVar is as follows:

$$\boldsymbol{\beta}^i = \boldsymbol{\beta}^{i-1} + \left(\mathbf{P}_{y,\rho}^*\right)^{\mathbf{T}} L'\left(\mathbf{x}_a'^{i-1}\right) + \left(\mathbf{P}_{y,\rho}^{\#}\right)^{\mathbf{T}} \mathbf{R}^{-1}\left[\mathbf{y}_{obs}' - L'\left(\mathbf{x}_a'^{i-1}\right)\right], \tag{21}$$

$$\mathbf{x}_a'^i = \mathbf{x}_a'^{i-1} + \mathbf{P}_{x,\rho}\left(\mathbf{P}_{y,\rho}^*\right)^{\mathbf{T}} L'\left(\mathbf{x}_a'^{i-1}\right) + \mathbf{P}_{x,\rho}\left(\mathbf{P}_{y,\rho}^{\#}\right)^{\mathbf{T}} \mathbf{R}^{-1}\left[\mathbf{y}_{obs}' - L'\left(\mathbf{x}_a'^{i-1}\right)\right], \tag{22}$$

and

$$\mathbf{P}_{x,\rho} = (\boldsymbol{\rho}_m < e > \mathbf{P}_x) = \left(\boldsymbol{\rho}_m \circ \mathbf{P}_{x,1}^*, \boldsymbol{\rho}_m \circ \mathbf{P}_{x,2}^*, \cdots, \boldsymbol{\rho}_m \circ \mathbf{P}_{x,N}^*\right), \tag{23}$$

$$\mathbf{P}_{y,\rho}^* = \left(\boldsymbol{\rho}_o < e > \mathbf{P}_y^*\right), \quad \mathbf{P}_{y,\rho}^{\#} = \left(\boldsymbol{\rho}_o < e > \mathbf{P}_y^{\#}\right). \tag{24}$$

Here, symbol $< e >$ is given in Eq. (23), symbol "$\circ$" is the Schür product, each column of $\mathbf{P}_{x,j}^* \in \mathbb{R}^{n_m \times r}$ ($j = 1, 2, \cdots, N$) is the same as the $j$th column of $\mathbf{P}_x$, $\boldsymbol{\rho}_m \in \mathbb{R}^{n_m \times r}$ is the decomposition matrix of the model grids spatial correlation matrix, and

$\boldsymbol{\rho}_o \in \mathbb{R}^{n_o \times r}$ is extracted from the decomposition matrix of the correlation matrix:

$$\mathbf{C}_{mo} \approx \boldsymbol{\rho}_m \boldsymbol{\rho}_o^{\mathbf{T}}. \tag{25}$$

$\mathbf{C}_{mo} \in \mathbb{R}^{n_m \times n_o}$ is the correlation matrix between the model grids and observation positions constructed by the following fifth-order piecewise rational function (Gaspari and Cohn, 1999):

$$\mathbf{C}_{mo}(i, j) = \mathbf{C}_0(d_{i,j} / d), \tag{26}$$

where $\mathbf{C}_0$ is defined as

$$\mathbf{C}_0(l) = \begin{cases} -\dfrac{1}{4}l^5 + \dfrac{1}{2}l^4 + \dfrac{5}{8}l^3 - \dfrac{5}{3}l^2 + 1, & 0 \leq l \leq 1 \\ \dfrac{1}{12}l^5 - \dfrac{1}{2}l^4 + \dfrac{5}{8}l^3 + \dfrac{5}{3}l^2 - 5l + 4 - \dfrac{2}{3}l^{-1}, & 1 < l \leq 2 \\ 0, & 2 < l \end{cases} \tag{27}$$

$l = \dfrac{d_{i,j}}{d}$, $d$ is the localization radii, $d_{i,j}$ is the spatial spherical distance between $i$th model grid and $j$th observation, $n_m$ is the model grid number, $n_o$ is the observation number, and $r$ is the number of selected truncation modes.



In the Tan-Tracker (v1) assimilation system, the optimization variables for different assimilation passes differ. In the $CO_2$ assimilation pass, the optimized state variable $\mathbf{x}$ is the $CO_2$ concentration $\mathbf{U}$, and $\mathbf{x}_a^{'}$ is the increase in the initial $CO_2$ concentration. IPs $\mathbf{P}_x$ are the initial perturbations of the $CO_2$ concentration, and OPs $\mathbf{P}_y$ are the perturbations of simulated $XCO_2$ within the 3-day window; $H_k$ is the observation operator of $XCO_2$ given in Eq. (31). For the flux assimilation pass,

state variable $\mathbf{x}$ is the scale factor $\boldsymbol{\lambda}$, and $\mathbf{x}_a^{'}$ is the increase in the scale factor within the window. IPs $\mathbf{P}_x$ and OPs $\mathbf{P}_y$ are the scale factor perturbations and simulated $XCO_2$ perturbations, respectively, within the 2-week window. At this point, the observation operator $H_k$ can be considered as a two-part chemistry transport model and the observation operator of the column-average concentration $XCO_2$.

## 2.3 Ensemble generation and update of the Tan-Tracker (v1) assimilation system

The NLS-4DVar assimilation algorithm is an ensemble-based algorithm that is used to approximate the analysis incremental solution space with the ensemble perturbation sample space. As such, the generation and update of the ensemble samples are essential for assimilation accuracy. According to the characteristics of $CO_2$ and the flux assimilation pass, we designed different sampling and updating methods for the Tan-Tracker (v1) assimilation system.

A historical moving sampling scheme (Wang et al., 2010; Tian et al., 2014) was used in the $CO_2$ assimilation pass to select

samples from a long-term historical $CO_2$ simulation, and a resampling scheme was used in the new assimilation window. The advantage of selecting samples from the historical simulation is that the appropriate sample size can be selected to ensure good results at a low computational cost. In this study, $N = 160$ was selected in the experiments to achieve a better assimilation effect.

To ensure better flux results and minimize computational cost, we chose an ensemble number of $N = 36$ in the flux

assimilation pass, integrating from the same initial $CO_2$ concentration within each window; all ensembles ran throughout the entire assimilation process. The ensemble generation scheme of the flux assimilation pass combines the history sampling and ensemble update. The historical sampling was applied to the initial window, and the $N = 36$ initial ensemble members were selected by a moving strategy. Ensemble samples of subsequent windows were obtained using the ensemble update given by the Local Ensemble Transform Kalman Filter (Hunt et al., 2007; Tian and Xie, 2012):

$$\mathbf{P}_x^a = \mathbf{P}_x \mathbf{T}, \tag{28}$$

where $\mathbf{P}_x^a$ represents the updated ensemble perturbations, and the transformation matrix $\mathbf{T}$ is given by

$$\mathbf{T} = \left[ (N-1)\mathbf{P}^* \right]^{(1/2)}, \tag{29}$$

with

$$\mathbf{P}^* = \left[ \mathbf{P}_y^{\mathbf{T}} \mathbf{R}^{-1} \mathbf{P}_y + (N-1)\mathbf{I} \right]^{-1}. \tag{30}$$





Equation (28) indicates that the updated ensemble perturbation $\mathbf{P}_x^a$ can be obtained from the initial perturbation $\mathbf{P}_x$ and a transformation matrix $\mathbf{T}$.

As the assimilation cycle progresses, the above ensemble update method usually reduces the dispersion of ensemble samples (Wang and Bishop, 2003), leading to an approximate distortion of the ensemble space $\mathbf{P}_x^a$ with respect to the solution space

$\mathbf{x}_a^{'}$; this ultimately causes the assimilation to fail. Therefore, we used an inflation factor $\sqrt{\eta}$ (see Zheng et al. (2013) for more details) with the ensemble perturbation $\mathbf{P}_x^a$, in which $\sqrt{\eta}\mathbf{P}_x^a$ maintained the dispersion of the ensemble samples; this is referred to as adaptive ensemble inflation.

## 3. Observing System Simulation Experiments

### 3.1 Model settings and observations

The Tan-Tracker (v1) carbon cycle data assimilation system is based on the global three-dimensional (3D) atmospheric chemistry model GEOS-Chem (version: v11-01, http://acmg.seas.harvard.edu/geos), driven by meteorological inputs of Modern-Era Retrospective analysis for Research and Applications (MERRA-2) from the GEOS of the National Aeronautics and Space Administration (NASA, United States) Global Modeling and Assimilation Office. The original GEOS-Chem $CO_2$ simulation was developed by Suntharalingam et al. (2004). A major update to the $CO_2$ simulation was completed by Nassar

et al. (2010). The latest update to the $CO_2$ simulation was developed by Nassar et al. (2013) and appears in GEOS-Chem v10-01, which was released in 2015. In the following experiments, we used the same spatiotemporal resolution: a horizontal resolution of $2° \times 2.5°$ (latitude $\times$ longitude), 47 vertical layers, a chemical time step of 20 min, a transmission time step of 10 min, and an output time of 3 h for the $CO_2$ concentration.

The fluxes used to drive GEOS-Chem for the $CO_2$ simulation were integrated and provided by the Harvard–NASA

Emissions Component (HEMCO) model (Keller et al., 2014). There are seven emission inputs from the following sources: fossil fuel, ocean exchange, terrestrial ecosystem fluxes, biomass burning, ships, aviation, and chemical oxidation. Fossil fuel emissions were acquired from the Open-source Data Inventory of Anthropogenic $CO_2$ (ODIAC) (Oda and Maksyutov, 2011) daily emissions data. Ocean exchange emissions were obtained from daily scaling data by Takahashi et al. (2009). Terrestrial ecosystem fluxes, specifically balanced biosphere exchange with a seasonal cycle but zero net annual uptake,

were taken from the hourly data provided by the Simple Biosphere (SBI3) model (Baker et al., 2006; Messerschmidt et al., 2013). Biomass burning emissions were obtained from the Global Fire Emissions Database v4 (GFED4) (Randerson et al., 2018) daily biomass burning data. Ship emissions were based on monthly scaling data from Endresen et al. (2007). Aviation emissions were derived from monthly scaling data (Olsen et al., 2013) from the Aviation Emissions Inventory Code (AEIC) (Simone et al., 2013). Sources of carbonaceous compound oxidation were taken from monthly data provided by Nassar et al.

30    (2010).



The observations used in the OSSEs are based on real OCO-2 satellite column-average concentration $XCO_2$ data (Crisp et al., 2017), data version v8r (https://disc.gsfc.nasa.gov/datasets/OCO2_L2_Lite_FP_V8r/summary; OCO-2 Science Team, 2017). From Connor's (2008) algorithm, we constructed an OCO-2 satellite $XCO_2$ observation operator, representing a projection from 3D atmospheric $CO_2$ concentrations to satellite column-average concentration:

$$X_{CO_2} = X_{CO_2,ap} + \mathbf{A} \cdot \left( U - U_{ap} \right), \tag{31}$$

where $X_{CO_2,ap}$ is the prior column-average concentration, $\mathbf{A}$ is the column-averaging kernel matrix, $U_{ap}$ is the prior $CO_2$ profile, and $U$ is the profile of the 3D atmospheric $CO_2$ concentration in each pressure layer of the prior $CO_2$ profile, used here as the interpolation result of the GEOS-Chem simulation profile.

When constructing the satellite observation $X_{CO_2,O}$, we retained the prior $CO_2$ profile, the prior column-average

concentration, the column-averaging kernel matrix, the pressure profile, quality control parameters, and time and position information. Only the column concentration value $X_{CO_2,Obs}$ and the uncertainty $X_{CO_2,un}$ from real observations were updated. The simulated truth profile $U_t$ was applied to Eq. (31) to obtain the simulated true column-average concentration:

$$X_{CO_2,t} = X_{CO_2,ap} + \mathbf{A} \cdot \left( U_t - U_{ap} \right). \tag{32}$$

By adding a normal distribution random error $X_{CO_2,err} \sim N\left(0, \mu\right)$ (Wang et al., 2010) instead of the observation uncertainty,

we were able to determine the simulated column-average concentration:

$$X_{CO_2,O} = X_{CO_2,t} + X_{CO_2,err}, \tag{33}$$

instead of $X_{CO_2,Obs}$, which was used as artificial observations in the OSSEs.

The artificial observations were controlled for quality to ensure that the OSSEs were reasonable and close to the actual situation. We set the quality-control parameter Warn level = 0 (representing the 50% best data) and used an observation –

background (O-B) 3σ quality control scheme for data culling. A comparison of artificial data before and after quality control in the first window (2 weeks) of the flux assimilation pass is shown in Figure 2. It is worth noting that Tan-Tracker (v1) does not use data thinning or regional average observations for assimilation, but instead applies the NLS-4DVar algorithm based on an efficient localization scheme (Zhang and Tian, 2018); this allows this tracking system to absorb large amounts of observations in a short period of time (about $10^5$ per window in the $CO_2$ assimilation pass and about $10^6$ per window in the

flux assimilation pass).

## 3.2 Observing system simulation experiments

The initial atmospheric $CO_2$ concentration represents the state of the atmospheric carbon pool at the initial time, which is important for the simulation of $CO_2$ concentration and flux inversion. All of the initial atmospheric $CO_2$ concentrations in the following experiments were from the Carbon-Tracker 2017 global $CO_2$ concentration (Peters et al., 2007;



https://www.esrl.noaa.gov/gmd/ccgg/carbontracker/), interpolated from a global resolution of $2° \times 3°$ (latitude $\times$ longitude), with 25 vertical layers, to the GEOS-Chem model grid resolution.

We designed a set of OSSEs as shown in Table 1. Experimental *True* represents the true simulation, starting from the initial $CO_2$ of the Carbon-Tracker global $CO_2$ at time 20151101 (for short: CT20151101), running from 20151101 to 20161231.

Forcing was driven by true fluxes: the terrestrial ecosystem flux of SIB3 in 2010 and the Takahashi ocean flux in 2010. Artificial observations $X_{CO_2,o}$ of *True* were constructed as discussed in Section 3.1. We also designed a background simulation control run (denoted as *Ctrl*), an assimilation experiment Tan-Tracker (v0) (denoted as *TT_v0*), and an assimilation experiment Tan-Tracker (v1) (denoted as *TT_v1*), with the same initial $CO_2$ CT20160101, the same running time from 20160101 to 20161231, and the same prior (background) fluxes: the terrestrial ecosystem flux of SIB3 in 2009

and the Takahashi ocean flux in 2009. The rest of the model settings remained the same as in *True*, with the difference being that *TT_v0* and *TT_v1* were assimilation experiments, assimilating artificial observations $X_{CO_2,o}$ of *True*.

The settings and parameters for *TT_v0* can be found in Tan-Tracker (Tian et al., 2014), where only observation data, model versions, and prior flux replacements were performed. A comparison of the parameter settings of *TT_v0* and *TT_v1* is shown in Table 2. After the sensitivity test, the localization radii of the $CO_2$ assimilation pass and flux assimilation pass were both

selected to be 2000 km, and the localization truncation modes numbers were $r_x = 50$ and $r_y = 30$ (see Zhang and Tian (2018) for details regarding the selection of localization-related parameters).

## 3.3 Analysis of results

### 3.3.1 $CO_2$ concentration

The $CO_2$ concentration reflects the state of the atmospheric carbon pool and can be used as a basic indicator for verification

in flux inversion. Here, we analyzed the $CO_2$ concentration results in detail from the time series and spatial distributions. We used the time series of the daily root-mean-square error (RMSE) and the time series of the mean deviations to characterize the deviations of *Ctrl*, *TT_v0*, and *TT_v1* from *True* (Fig. 3). Overall, the indicators in Figure 3 showed that the results after the assimilation were better than the background results.

The daily RMSE of XCO$_2$ between the simulation/assimilation and artificial observations (Fig. 3a), representing the change

in column-average concentration at the observed position, provides a comparison between O-B and the difference between observations and assimilation (O-A) to explain the effectiveness of the assimilation. The results in Figure 3a showed that the two versions of the Tan-Tracker carbon cycle data assimilation system effectively absorbed observations for flux optimization, with the *TT_v1* showing slightly better performance than *TT_v0* and superior performance with respect to that of *Ctrl*.

Daily RMSE (Fig. 3b) and the daily mean bias (Fig. 3c) of the atmospheric 3D $CO_2$ concentration between the simulation/assimilation results and *True* reflect the changes in the atmospheric carbon pool. Figure 3b shows the deviation of



the simulation/assimilation results from *True*. The deviation between *Ctrl* and *True* decreased from 1.4 to 0.4 ppm at the initial time from January to February, and remained low (0.4 ppm) from March to June; this showed that the initial concentration deviation was reduced gradually, which could be considered as a model spin-up process. The deviation increased from July to September from 0.6 to 1.0 ppm, which indicated that there was a large deviation between the prior flux and the true flux in the Northern Hemisphere growth season. Finally, the deviation from October to December fell back to 0.3–0.4 ppm, indicating a decrease in the deviation between the prior flux and the true flux in the non-growth season of the Northern Hemisphere. The deviation between *TT_v1* and *True* decreased from 1.4 to 0.2 ppm at the initial time from January to February, maintaining a lower value of 0.2 ppm from March to June. After a slight increase to 0.2–0.6 ppm from July to September, the deviation between *TT_v1* and *True* finally fell back to 0.2 ppm from October to December.

Figure 3c shows the daily mean bias between the simulation/assimilation results and *True*. The daily mean bias of *TT_v1* dropped rapidly from −0.4 to 0 ppm and then remained low (-0.05 to 0.05 ppm); this performance was superior to that of *Ctrl*, which showed a larger bias amplitude. Thus, *TT_v1* exhibited a faster spin-up convergence speed and a smaller deviation over the entire simulation time than *Ctrl*; these improvements were attributed to an adjustment in the optimized flux. The effect of the initial $CO_2$ optimized by the $CO_2$ assimilation pass occurred only at the initial time of each window, thus only a small adjustment to the state of the atmospheric carbon pool, and mainly served to improve the accuracy of the optimized flux. This was achieved given the good continuity of the $CO_2$ results (Figs. 3b and 3c). The results of *TT_v0* were better than those of *Ctrl* but slightly inferior to those of *TT_v1*.

Figure 4 shows the spatial deviation between the simulation/assimilation results and *True* based on the RMSE spatial distribution of the vertical-averaged $CO_2$ concentration grid time series. Figure 4a displays the RMSE spatial distribution between *Ctrl* and *True*. Large values over land appeared in Western Siberia (1.0–1.2 ppm) and Eastern Siberia, Eastern Central Asia, Eastern North America, and Central South America (0.8–1.0 ppm). Large values over the ocean appeared in the Northern Hemisphere, with an increasing bias trend from the Southern Hemisphere to the Northern Hemisphere (0.2–0.5 ppm). The results of *TT_v1* were better than those of *Ctrl*, with a large bias over land of 0.3–0.5 ppm; the increasing bias trend over the ocean was lower at 0.1–0.2 ppm. The results of *TT_v0* were better than those of *Ctrl* and slightly inferior to those of *TT_v1*.

### 3.3.2 Flux

In real assimilation experiments, $CO_2$ concentration results can be used as the main objective indicator of flux evaluation due to the lack of a real flux. However, in OSSEs, we can analyze the prior flux quantitatively and optimize the flux and real flux to give the most direct judgment. Below we present a detailed analysis of flux using time series, annual total amounts, and regional distributions.

Figure 5 shows the time series of the simulation/assimilation results of the monthly global total ecosystem, the ocean flux, and their deviations from *True*. Notably, similar to the spin-up process of the numerical model simulation, the first 4 months corresponded to the spin-up process of the flux assimilation pass. During the early stages of the spin-up phase (from January





to February), a larger portion of the optimized flux increment was used to adjust the initial $CO_2$ concentration deviation from the true simulation. As a result, the deviation between the optimized flux and the true flux was larger than the prior value (Fig. 5b); however, the $CO_2$ concentration deviation continued to decrease (Fig. 3b). As the assimilation progressed, the concentration deviation became more stable. At this time point, the uncertainties in $CO_2$ concentration and flux could not be

distinguished; as such, the assimilation continued to run, allowing for adjustments to the flux and concentration. Finally, the deviations caused during the corresponding flux and concentration optimization processes were minimized. Here, we mainly discuss the flux results from May to December after reaching equilibrium.

The prior flux (*Ctrl*) was in good agreement with the true flux (*True*) (Fig. 5a). Additionally, a significant seasonal cycle was evident (Fig. 5a). April to September is the growing season of the Northern Hemisphere, when the total flux of the global

terrestrial ecosystem and oceans is negative, reaching its lowest value in July and August. From October to March, corresponding to the non-growth season in the Northern Hemisphere, the global flux was positive, and there was no obvious monthly change. The main deviation of the prior flux (*Ctrl* in Fig. 5b) appeared in the Northern Hemisphere growing season from June to August, reaching $-4.0$ PgC yr$^{-1}$. In addition, there was a significant deviation of about 0.2 PgC yr$^{-1}$ during the non-growth season of the Northern Hemisphere from October to December. The *TT_v1* optimized flux of the dual-pass

system showed significant improvement over *Ctrl*. The deviation was reduced to 0.0 PgC yr$^{-1}$ from June to August, and the deviation decreased to 0.1 PgC yr$^{-1}$ from October to December (Fig. 5b). The results from *TT_v0* were better than those of *Ctrl*, but slightly inferior to those of *TT_v1*. Table 3 and Figure 6 show the assimilation/simulation deviations of the terrestrial ecosystem flux, ocean flux, and global total flux from *True* from May to December. Compared with *Ctrl*, the results of *TT_v1* were better optimized for the terrestrial ecosystem flux and slightly improved for the ocean flux. In addition,

the results of *TT_v0* were better than those of *Ctrl*, but slightly inferior to those of *TT_v1* for the terrestrial ecosystem flux and slightly superior to those of *TT_v1* for ocean flux.

We used the TransCom "super-regions" (Gurney et al., 2002) to calculate the regional total flux. Figure 7 shows the flux results of 11 land regions and the deviation from *True*. The results of *TT_v1* had a positive effect on each region relative to the prior flux of *Ctrl*, with significant improvements in the mid-to-high latitudes of North America, Europe, and Eurasia, and

the mid-latitudes of South America and Australia. The results in the equatorial region of South America and Asia did not show significant improvements. The prior flux in Africa was close to the true value; an increase was not obvious in the data. *TT_v0* showed slightly improved results compared with *Ctrl*, but both were inferior to the performance of *TT_v1*.

### 3.3.3 Sensitivity experiments

The parameters of the carbon cycle data assimilation system Tan-Tracker (v1) are listed in Table 2. The main parameters are

the assimilation window length and the maximum NLS-4DVar assimilation iteration number of the flux assimilation pass, as described below.

The flux assimilation pass window length determines the influence of the initial $CO_2$ concentration and the time of transmission, thus affecting the flux inversion. The sensitivity experiments of the assimilation window were used to select a





window length of 7 days (denoted as *v1_07*), 14 days (denoted as *v1_14*), or 30 days (denoted as *v1_30*); the other *TT_v1*
parameters remained unchanged. The flux and concentration results are shown in Figure 8. From the time series of the total
flux (Fig. 8a,) it could be concluded that the assimilation experiments of all three windows had positive effects; however, the
assimilation results of *v1_14* were better than those of *v1_07*, which was better than those of *v1_30*. The $CO_2$ concentration
results (Fig. 8c) showed that the assimilation experiments of all three windows had positive effects. The assimilation results
of *v1_07* were roughly equivalent to those of *v1_14*, both of which were better than those of *v1_30*. Thus, flux assimilation is
sensitive to the length of the assimilation window. The window of the appropriate length (14 days) had a small initial $CO_2$
concentration deviation, the appropriate integration time, and was closest to the OCO-2 satellite 16-day regression period,
i.e., it was possible to absorb more observations to obtain good flux inversion results.
As the maximum NLS-4DVar iteration number increases, the assimilation results tend to converge, especially for solving the
problem of high nonlinear systems. However, the computational cost increases with the number of iterations. The sensitivity
experiments of the maximum NLS-4DVar iteration number selected one (*Imax = 1*), two (*Imax = 2*), and three (*Imax = 3*)
iterations, with the remaining parameters retaining the values of *TT_v1*. The resulting flux and concentration results are
shown in Figure 9. The time series of the monthly total flux (Fig. 9a) and the $CO_2$ concentration (Fig. 9c) results showed that
the assimilation results improved and tended to converge quickly as the number of maximum NLS-4DVar iterations
increased. Considering the computational cost, we chose three maximum NLS-4DVar iterations as the final solution.

## 4. Conclusion

We designed a new version of a carbon cycle data assimilation system, Tan-Tracker (v1), based on the atmospheric chemical
transport model GEOS-Chem and an advanced NLS-4DVar data assimilation algorithm. Using a dual-pass assimilation
framework consisting of a $CO_2$ assimilation pass and a flux assimilation pass, we assimilated atmospheric $CO_2$ observations
to obtain an optimized representation of the surface carbon flux. Compared with the joint assimilation system Tan-Tracker
(v0), the dual-pass assimilation system Tan-Tracker (v1) innovatively uses a dual-pass assimilation framework to
successively optimize $CO_2$ concentration and surface carbon flux in different assimilation passes. Optimization of the $CO_2$
concentration uses a shorter assimilation window to reduce the effects of background flux for a more accurate initial $CO_2$
concentration measurement. Flux optimization uses a longer assimilation window, allowing the system to absorb enough
observations to optimize the flux while reducing the effects of the initial $CO_2$ concentration deviation, resulting in more
accurate surface flux estimates.

We designed a set of OSSEs based on OCO-2 satellite data, which we compared with the Tan-Tracker (v0) joint assimilation
system. The Tan-Tracker (v1) performance was superior to that of Tan-Tracker (v0) in resolving the $CO_2$ concentration and
surface flux estimates, and was far better than direct background simulations. Thus, the dual-pass assimilation strategy offers
an advantage in satellite carbon cycle data assimilation. The results of the sensitivity experiment of window length and

maximum NLS-4DVar assimilation iterations showed that the appropriate window length (14 days) and a greater number of iterations (three), as permitted by the computational cost, provides better assimilation results.

Future work will focus on multi-satellite (e.g., OCO-2, GOSAT, and Tan-Sat) observations for long-term sequence assimilation, regional high-resolution nested assimilation, and analyses used to distinguish between anthropogenic and natural sources.

**Author contribution**

X. Tian: conception and design, data analysis and interpretation, manuscript writing, final approval of manuscript; R. Han: model simulation, data analysis and interpretation, manuscript writing.

**Code and data availability**

Initial global $CO_2$ concentrations are from Carbon-Tracker CT2017 results provided by NOAA ESRL, Boulder, Colorado, USA from the website at http://carbontracker.noaa.gov. Column-average concentration XCO2 data were produced by the OCO-2 project at the Jet Propulsion Laboratory, California Institute of Technology, and obtained from the OCO-2 data archive maintained at the NASA Goddard Earth Science Data and Information Services Center: data version v8r, https://disc.gsfc.nasa.gov/datasets/OCO2_L2_Lite_FP_V8r/summary. Global three-dimensional (3D) atmospheric chemistry model GEOS-Chem version: v11-01 is available at: http://acmg.seas.harvard.edu/geos. The code of the assimilation algorithm NLS-4DVar and POD-4DVar used in the system was originally from Xiangjun Tian and can be free and anonymous accessed at: https://doi.org/10.5281/zenodo.2677887.

**Acknowledgement**

The work was partially supported by the National Key Research and Development Program of China(2016YFA0600203), the Natural Science Foundation of China (41575100) and the Key Research Program of Frontier Sciences, Chinese Academy of Sciences (CAS)(QYZDY-SSW-DQC012).

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

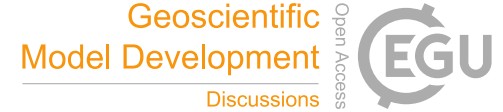



**Table 1. Experimental setup of the observing system simulation experiments (OSSEs). SIB3 and Takahashi flux are as described in Section 3.1; the remaining flux in each experiment is the same.**

| Name | Running time | Initial $CO_2$ | Flux *bio* | Flux *oce* |
|------|--------------|----------------|------------|------------|
| *True* | 20151101~20161231 | CT 20151101 | SIB3 2010 | Takahashi 2010 |
| *Ctrl* | 20160101~20161231 | CT 20160101 | SIB3 2009 | Takahashi 2009 |
| *TT_v0* | | | | |
| *TT_v1* | | | | |

5  **Table 2. Selection of assimilation parameters (parameters for *TT_v1* divided into the $CO_2$ assimilation pass and the flux assimilation pass).**

| Name | Window length(days) | Lag window (days) | Localization radius(km) | Localization parameters($r_x$, $r_y$) | Iteration times |
|------|---------------------|-------------------|-------------------------|---------------------------------------|-----------------|
| *TT_v0* | 7 | 35(5 weeks) | 900 | | 1 |
| $CO_2$ pass | 3 | | 2000 | 50,30 | 3 |
| Flux pass | 14(2 weeks) | | 2000 | 50,30 | 3 |

**Table 3. Total flux from May to December and its deviation from *True* (unit: PgC yr$^{-1}$).**

| | *True* | *Ctrl* | *TT_v0* | *TT_v1* |
|------|--------|--------|---------|---------|
| Ocean | -2.03069 | -1.87451 | -1.88218 | -1.87805 |
| land ecosystem | -4.09202 | -3.96772 | -4.04544 | -4.10416 |
| total | -6.12273 | -5.84224 | -5.92764 | -5.9822 |
| | | *Ctrl* | *TT_v0* | *TT_v1* |
| Ocean | | 0.15618 | 0.148511 | 0.152643 |
| land ecosystem | | 0.124296 | 0.046578 | -0.01214 |
| total | | 0.280489 | 0.195089 | 0.140531 |





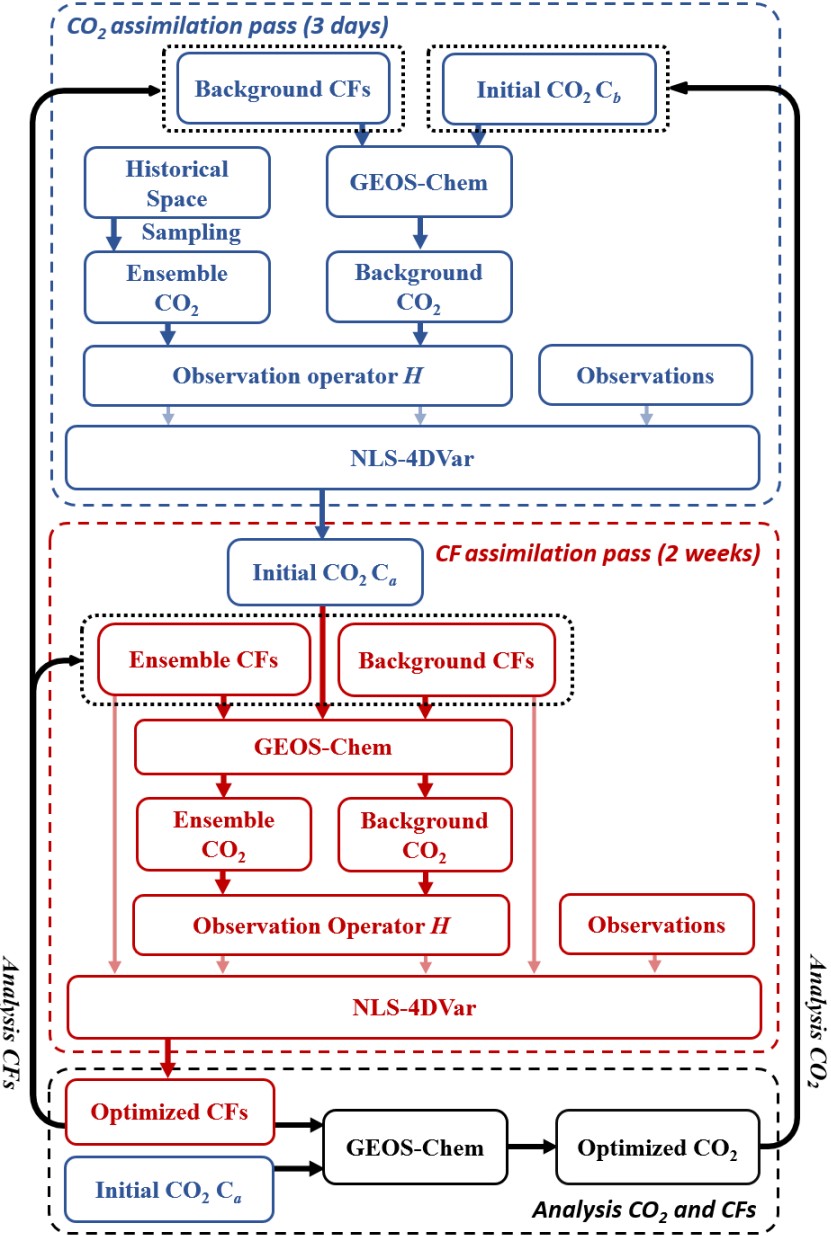

**Figure 1. Dual-pass Tan-Tracker (v1) assimilation system framework.**



**Figure 2. Spatial distribution of artificial observations $X_{CO_2,O}$ before and after quality control in the first window of the flux assimilation pass.**





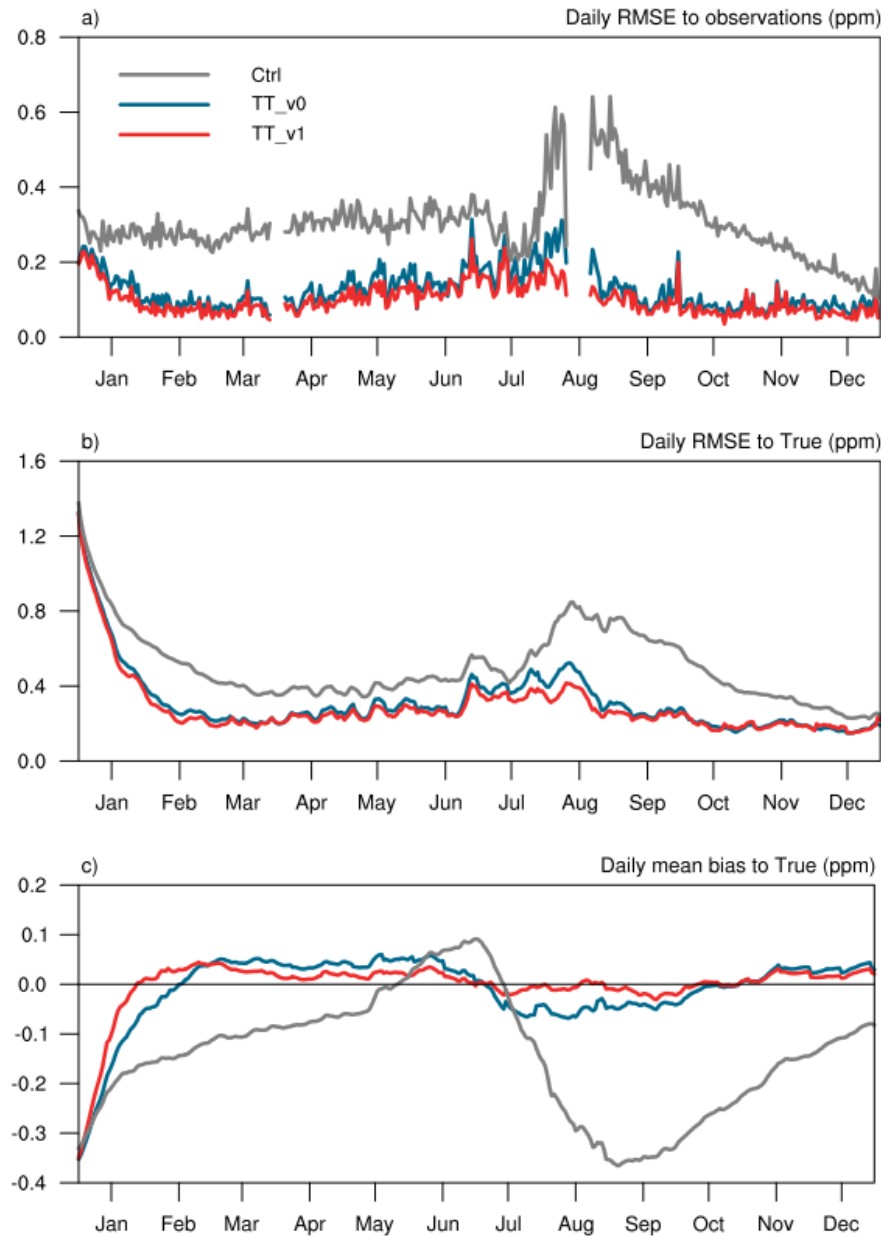

**Figure 3. CO₂ assimilation results: a. daily root-mean-square error (RMSE) between assimilation/simulation and artificial observations** $X_{CO_2,O}$ **; b. daily RMSE between assimilation/simulation and *True*; c. daily mean bias between assimilation/simulation and *True*.**



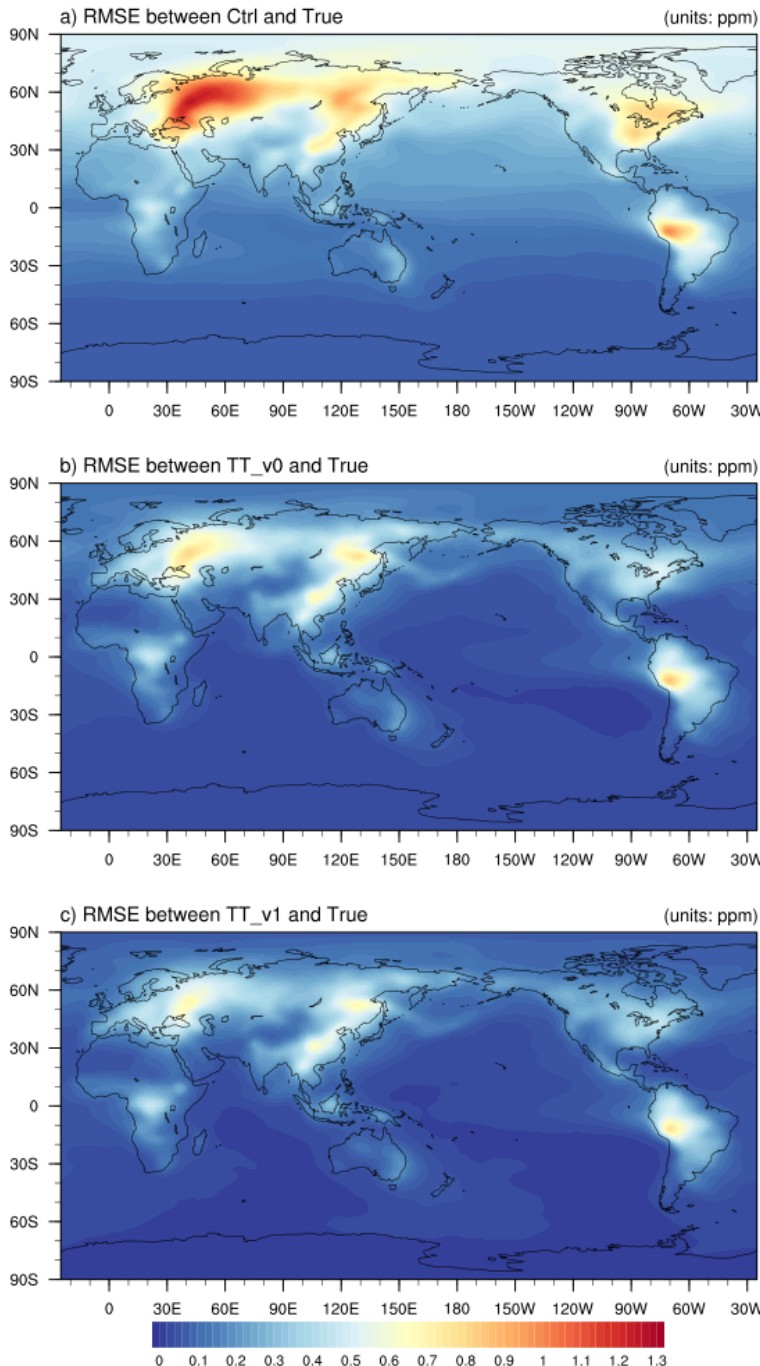

**Figure 4. Root-mean-square error (RMSE) spatial distribution of vertical-averaged CO₂ concentration grid time series. RMSE between a.** *Ctrl* **and** *True*; **b.** *TT_v0* **and** *True*; **c.** *TT_v1* **and** *True*.





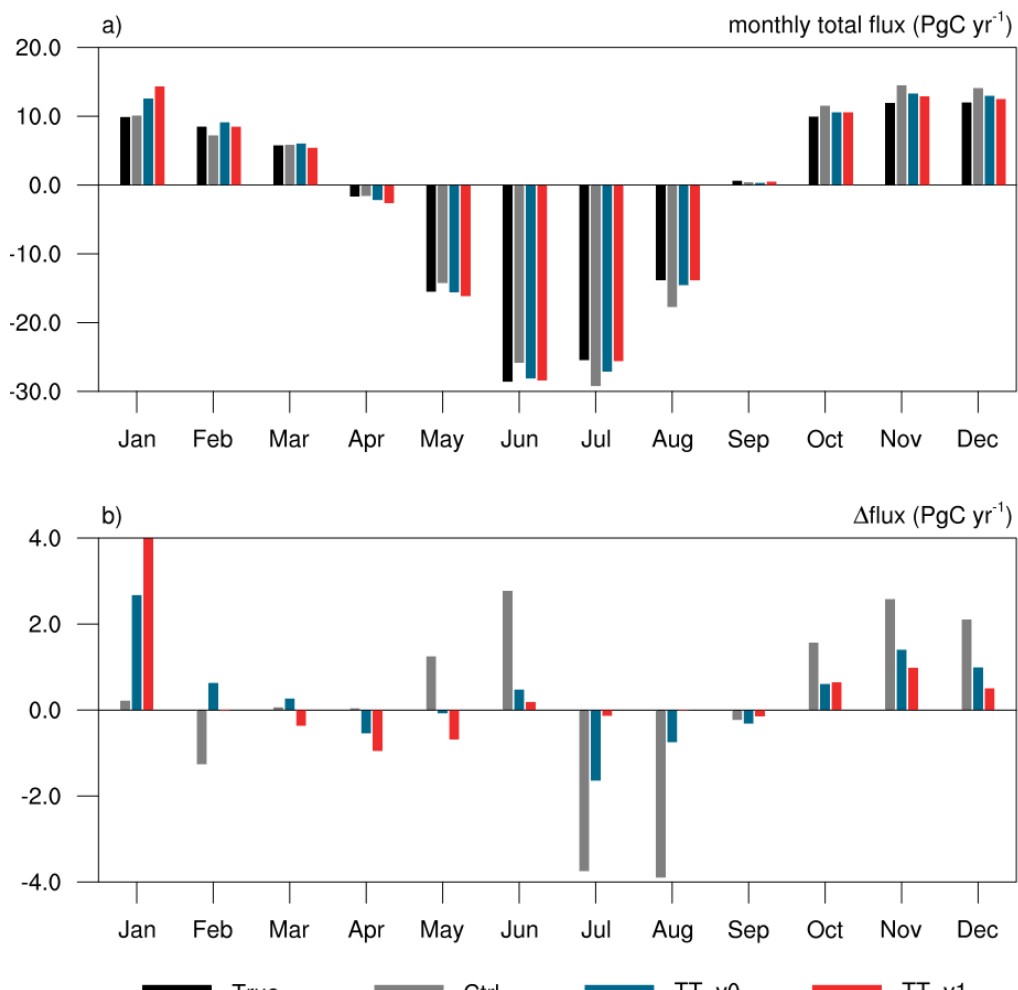

**Figure 5. Time series simulation/assimilation results of the monthly global total ecosystem and ocean flux and their deviation from the truth *True*: a. monthly total flux; b. monthly delta flux.**





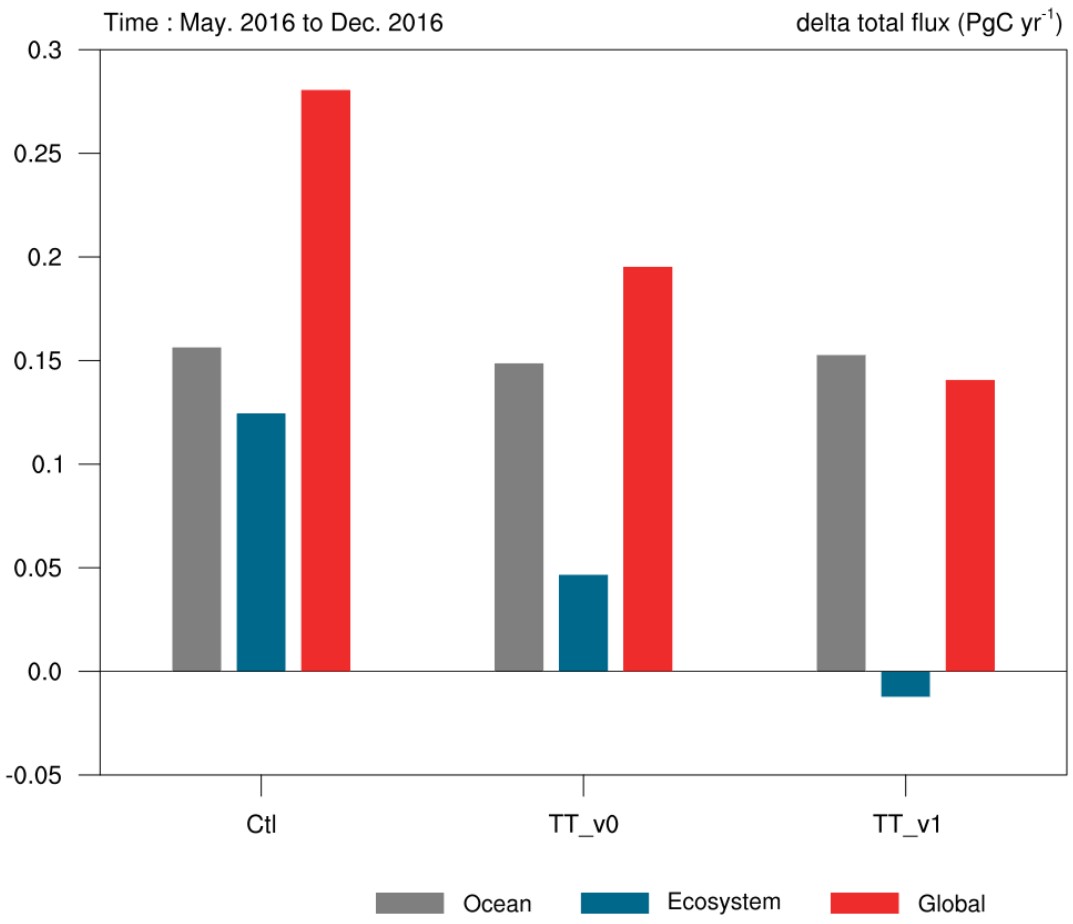

**Figure 6. Total flux from May to December and its deviation from *True*.**







**Figure 7. Monthly total flux of 11 land regions of TransCom "super-regions" and its deviation from *True*: a. flux of each region; b. deviation from *True*.**





**Figure 8. Window length sensitivity experiment results: a. monthly total flux; b. monthly total flux deviation; c. daily root-mean-square error (RMSE) of CO₂ concentration between the simulation/assimilation results and *True*.**

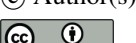



**Figure 9. Maximum Nonlinear Least Squares Four-dimensional Variational Data Assimilation algorithm (NLS-4DVar) iteration sensitivity experimental results: a. monthly total flux; b. monthly total flux deviation; c. daily root-mean-square error (RMSE) of CO₂ concentration between the simulation/assimilation results and *True*.**