# Peer review of "A dual-pass carbon cycle data assimilation system to estimate surface CO2 fluxes and 3D atmospheric CO2 concentrations from spaceborne measurements of atmospheric CO2"

_Geoscientific Model Development, 2019_

## Short Comment (SC1) · 17 May 2019

Dear authors,

in my role as Executive editor of GMD, I would like to bring to your attention our Editorial version 1.1:

http://www.geosci-model-dev.net/8/3487/2015/gmd-8-3487-2015.html

This highlights some requirements of papers published in GMD, which is also available

on the GMD website in the 'Manuscript Types' section:

http://www.geoscientific-model-development.net/submission/manuscript_types.html

In particular, please note that for your paper, the following requirements have not been met in the Discussions paper:

- "The main paper must give the model name and version number (or other unique identifier) in the title."

In order to simplify reference to your developments, please add a name (and/or an acronym) and a version number of the published data assimilation system in the title of your article in your revised submission to GMD.

Yours,

Astrid Kerkweg

---

## Author Comment (AC1) · 20 May 2019

Dear Editor:

Thank you very much for your comment and reminder, we really appreciate it. We will make a revision based on your valuable comment:

"In order to simplify reference to your developments, please add a name (and/or an acronym) and a version number of the published data assimilation system in the title of your article in your revised submission to GMD."

We will rename our title: "A dual-pass carbon cycle data assimilation system Tan-Tracker (v1) to estimate surface CO2 fluxes and 3D atmospheric CO2 concentrations from spaceborne measurements of atmospheric CO2" in our revised submission to GMD.

Best regards,

Yours sincerely,

Xiangjun Tian and Rui Han

- - - - - - - - - - - - - - - - - - - - - - - - - - - - - - - - - - - - - - - - - - - - - - - -

International Center for Climate and Environment Sciences (ICCES), Institute of Atmospheric Physics (IAP), Chinese Academy of Sciences (CAS), Beijing 100029, China

---

## Referee Comment (RC1) · Anonymous Referee #1 · 28 Jun 2019

The manuscript "A dual-pass carbon cycle data assimilation system to estimate surface CO2 fluxes and 3D atmospheric CO2 concentrations from spaceborne measurements of atmospheric CO2" by Han and Tian discusses a two step global CO2 natural flux inversion approach applied sequentially to relatively short (14-day) inversion cycles partitioning the full period of analysis (1 year). For each inversion cycle, the first step consists in optimizing the CO2 initial condition, and the second step consists in optimizing the CO2 fluxes. The system seems to strongly rely on the choice of a shorter assimilation window (at the beginning of the inversion cycle) for the first step, the initial

condition being constrained by a subset of the observations, while the fluxes are constrained using all observations of the inversion cycle in the second step (for which the assimilation window is the inversion cycle).

The authors attempt at demonstrating the advantages of such an approach using OSSEs with the assimilation of pseudo OCO-2 data, and comparing the results of Tan-Tracker(v1) (that uses such an approach and the NLS-4DVar system) to that of Tan-Tracker (v0) (which uses the POD-4DVar system and which optimizes the CO2 initial condition and fluxes simultaneously for each inversion cycle, without splitting the inversion into two steps).

I see severe issues in this study, in particular:

- regarding the theory: I can not understand how the split of the inversion cycle into 2 steps ("passes") could be an improvement. It raises theoretical issues, at least regarding the assimilation of some data twice. And it should hamper the proper distinction between the errors from the prior initial conditions and from the prior fluxes. The need to use a shorter assimilation window for the first pass is an indication of this limitation. Controlling both the CO2 initial condition and fluxes together, using the transport model and the prior uncertainties to drive the balance between corrections to the initial condition and to the surface fluxes, should lead to more robust results and is much more satisfying in terms of theory. The "dual pass" could be seen as a pragmatic way of controlling "manually" this balance (by playing with the length of the assimilation window for the first "pass"), but refining the set-up of the prior uncertainties in the initial condition and in the fluxes is a much proper way for such a control.

- in practice: my understanding is that the comparison between Tan-Tracker(v1) and Tan-Tracker(v0) in section 3 is completely biased. For the direct comparison in section 3.3.1 and 3.3.2, TTv1 uses 14-day inversion cycles while TTv0 uses 7-day inversion cycles, and, more critically, TTv1 uses 3 iterations for the minimization of the cost function, while TTv0 uses 1 iteration only (they also use different localization radius

and in practice, different systems asking for different parameters). Therefore, there is no reason to think that this comparison says something about the "dual pass" approach itself. Actually, TTv0 seems to provide results that are extremely similar to that of TTv1 using 1 iteration and 14-day inversion cycles (see Figure 5b vs.Figure 9b ) ! One could even assume that it provides better results than TTv1 using 1 iteration and 7-day inversion cycles (since results are better with 14-day cycles than with 7-day cycles for TTv1), not speaking about using 2000 km localization radius. My understanding is thus that the authors have misinterpreted their experiments and results.

The use of very short inversion cycles (here 14-days) exacerbates the problem of the corrections to initial conditions. For CO2 inversions, the use of short inversion cycles can hardly be seen as an advantage. Most of state of the art systems, especially global ones, use very long inversion cycles (1 year and more) to avoid breaking the link between uncertainties in the fluxes in an area and the errors at a remote location a long time later (which have to be solved for in the initial condition when using short inversion cycles while the target of inversion is a better estimate of the fluxes). I guess that the ensemble approach explains the need for short inversion cycles and maybe why results with 14-day inversion cycles are better than with 30-day inversion cycles. However, the manuscript does not attempt at explaining it.

The actual inversion system (i.e. the NLS-4DVar system, over which lies the TanTracker (v1) framework, and which is the actual code proposed in the "code and data availability" section) has already been detailed in past publications involving the second author. The section 2.2 is just the duplication of material from Tian et al. (2018), Zhang and Tian (2017), Tian and Feng (2015) and even in Tian et al. (2011). It thus cannot be a strong topic of this new manuscript, nor the overall changes from Tan-Tracker v0 to Tan-Tracker v1 i.e. from POD-4DVar with a single "pass" to NLS-4DVar with a "dual pass". Regarding the specific analysis of this paper on this NLS-4DVar system, I see in Figure 9 that the optimal number of iterations found for the minimization of the cost function, when testing 1,2 and 3 iterations is 3. But 3 is still much smaller to the typical number

of iterations usually used for such minimizations, and the experiments and analysis of this paper do not show that the minimization has converged after 3 iterations (Figure 9b even imply the opposite).

My opinion is thus this manuscript should be rejected.

It is important to note that the authors forget to say that their system is a global inversion system (and even to say that it inverts natural CO2 fluxes, until the details of the equations clarifies it) which should strongly influence the way the problem of the initial condition should be tackled, the choice of the inversion cycles and of the data assimilation windows, the size of the ensembles and the number of iterations used for the inversions... in a more general way, the authors ignore the influence of the specific framework of their inversions -domain, resolution, data assimilated- on their results and on their choices of values for the inversion parameters.

I add, without entering into too much details about it, that the quality of the text is not sufficient for a scientific publication. The abstract already gives a good illustration of the confusing way with which this manuscript is written. The authors insert a lot of technical jargon from the data assimilation community, but they actually misuse many of the corresponding terms (few examples: "surface flux inversion measurement", "flux assimilation" to speak about flux inversion assimilating CO2 data, the alternative use of "background" or "prior" to speak about the same thing, "one-step iteration", "atmospheric chemical transmission mode", "ensemble-based hybrid assimilation algorithm"...), leading to meaningless of confusing sentences. The abusive use of words and mathematical notations that seem more complicated than needed (or that they just forget to define, such as Py in eq 15) severely hampers the clarity of the paper. Most of the introduction sounds like a random sampling of references to past inversions, with meaningless comments (like "the surface carbon flux inversion method, obtained by combining model and atmospheric CO2 information, has made great progress in carbon cycle data assimilation", "For example, CarbonTracker is a well-designedÂăcarbon assimilation system", "Basu et al. showed that satellite data

provided an effective constraint for surface carbon source-sink inversion"...). It hardly provides clues about the specific topic of this paper. The analyzes in section 3.3 lack of depth and of hindsight on the significance and scope of the OSSEs and of the conclusions.

---

## Author Comment (AC2) · 17 Jul 2019

Responses to Anonymous Referee #1

We would like to thank the reviewer for carefully reading our paper and for all the valuable questions/comments/suggestions. We have thoroughly studied the reports and made a minor revision to our paper by incorporating all suggestions given in the reports. Below we give our itemized responses to all the comments.

"- regarding the theory: I can not understand how the split of the inversion cycle into 2

steps ("passes") could be an improvement. It raises theoretical issues, at least regarding the assimilation of some data twice. And it should hamper the proper distinction between the errors from the prior initial conditions and from the prior fluxes. The need to use a shorter assimilation window for the first pass is an indication of this limitation. Controlling both the CO2 initial condition and fluxes together, using the transport model and the prior uncertainties to drive the balance between corrections to the initial condition and to the surface fluxes, should lead to more robust results and is much more satisfying in terms of theory. The "dual pass" could be seen as a pragmatic way of controlling "manually" this balance (by playing with the length of the assimilation window for the first "pass"), but refining the set-up of the prior uncertainties in the initial condition and in the fluxes is a much proper way for such a control."

Response

The dual-pass data assimilation strategy is well developed and used in earth science, (Moradkhani et al., 2005a, 2005b; Vrugt et al., 2005; Xu et al., 2015); It is well known that the dual-pass data assimilation strategy is absolutely not a reuse of observation because that the observations are used in the two assimilation passes to assimilate different variables separately. The proposed dual-pass assimilation strategy is really an improvement in carbon cycle data assimilation. This novel dual-pass strategy (with different length windows) is specially proposed to give the roper distinction between the errors from the initial conditions and from the background fluxes: For a certain assimilation cycle, the simulated CO2 concentration errors originated from both the initial CO2 and the background flux errors. These errors entangled with the model evolution, which is indeed difficult for us to optimize the CO2 concentrations and fluxes altogether. Fortunately, in the proposed dual-pass assimilation strategy, a shorter 3-days CO2 assimilation pass is first adopted to optimize the initial CO2 concentrations. The optimized initial CO2 concentrations are then used for the following flux assimilation pass, which is just a proper distinction between the errors from the initial conditions and the fluxes (For more details, please see section 2.1 and Fig.1 of our manuscript).

In the ensemble-based Tan-Tracker system, the uncertainties are described by the ensembles in the NLS-4DVar, which are further optimized with the ensembles update (see 2.3 Ensemble generation and update of the Tan-Tracker (v1) assimilation system). In a summary, dual-pass presents a proper way controlling both CO2 initial condition and flux successively, using observations and transport model driven by refined prior uncertain. This is more consistent with the theory and is indeed an improvement of the carbon cycle data assimilation community. To be clear, we add the above description in 4. Discussion. (see the supplement to this comment)

"- in practice: my understanding is that the comparison between Tan-Tracker(v1) and Tan-Tracker(v0) in section 3 is completely biased. For the direct comparison in section 3.3.1 and 3.3.2, TTv1 uses 14-day inversion cycles while TTv0 uses 7-day inversion cycles, and, more critically, TTv1 uses 3 iterations for the minimization of the cost function, while TTv0 uses 1 iteration only (they also use different localization radius and in practice, different systems asking for different parameters). Therefore, there is no reason to think that this comparison says something about the "dual pass" approach itself. Actually, TTv0 seems to provide results that are extremely similar to that of TTv1 using 1 iteration and 14-day inversion cycles (see Figure 5b vs. Figure 9b )! One could even assume that it provides better results than TTv1 using 1 iteration and 7-day inversion cycles (since results are better with 14-day cycles than with 7-day cycles for TTv1), not speaking about using 2000 km localization radius. My understanding is thus that the authors have misinterpreted their experiments and results."

Response

Actually, the comparisons between Tan-Tracker (v0) and Tan-Tracker (v1) are performed with their respective optimal parameters. The same assimilation parameters have different effects and sensitivities in different systems due to different assimilation framework, assimilation algorithm, and different ensemble generation and update methods. To be clear, we add "Note that the comparisons between TT_v1 and TT_v0 are performed with their respective optimal parameters." on page 11 line 7. As complementary, we conducted another group of experiments with the 7-day cycles for the maximum NLS-4DVar iteration number Imax=1 and Imax=3 (see the following figure fig. 1): the TT_v0 performance is not improved with the increasing Imax and the TT_v1 with Imax=3 is the best one among all the experiments.

"The use of very short inversion cycles (here 14-days) exacerbates the problem of the corrections to initial conditions. For CO2 inversions, the use of short inversion cycles can hardly be seen as an advantage. Most of state of the art systems, especially global ones, use very long inversion cycles (1 year and more) to avoid breaking the link between uncertainties in the fluxes in an area and the errors at a remote location a long time later (which have to be solved for in the initial condition when using short inversion cycles while the target of inversion is a better estimate of the fluxes). I guess that the ensemble approach explains the need for short inversion cycles and maybe why results with 14-day inversion cycles are better than with 30-day inversion cycles. However, the manuscript does not attempt at explaining it."

Response

14-days window length is very close to those adopted by other published inversions systems, such as the one week length of Carbon-Tracker (Peters et al., 2007), the 8-days length of Chevallier et al. (2011) and the one-month length of Basu et al. (2013) also the 7-days length of Tan-Tracker (v0) (Tian et al., 2014). Similarly, we also utilize the sensitivity experiments (shown in Fig.8) to determine the optimal assimilation window-length. To be clear, we add "Note that, 14-days flux assimilation pass window length is close to those adopted by some other published inversion systems, such as the one week length of Carbon-Tracker (Peters et al., 2007), the one-month length of Basu et al. (2013) also the 7-days length of Tan-Tracker (v0) (Tian et al., 2014)." on page 14 line 3.

"The actual inversion system (i.e. the NLS-4DVar system, over which lies the TanTracker (v1) framework, and which is the actual code proposed in the "code and

data availability" section) has already been detailed in past publications involving the second author. The section 2.2 is just the duplication of material from Tian et al. (2018), Zhang and Tian (2017), Tian and Feng (2015) and even in Tian et al. (2011). It thus cannot be a strong topic of this new manuscript, nor the overall changes from Tan-Tracker v0 to Tan- Tracker v1 i.e. from POD-4DVar with a single "pass" to NLS-4DVar with a "dual pass"."

Response

The NLS-4DVar is an indispensable part of the Tan-Tracker system and some necessary descriptions of this method is needed for the self-consistency of Tan-Tracker, which is also required by the our topical editor: "Additionally, I would suggest to the authors to slightly improve the description of the "assimilation method" as some details seem to be missing for a complete evaluation of the method without reading the two papers, Tian and Feng (2015) and Tian et al. (2018)". As stated above, the proposed dual-pass assimilation strategy is really an improvement in carbon cycle data assimilation. This novel dual-pass strategy (with different length windows) is specially proposed to give the roper distinction between the errors from the initial conditions and the prior fluxes. In additional, the NLS-4DVar algorithm with efficient localization scheme used in Tan-Tracker leads to high assimilation precision with lower computational costs, which is certainly a great improvement to a data assimilation system.  

"Regarding the specific analysis of this paper on this NLS-4DVar system, I see in Figure 9 that the optimal number of iterations found for the minimization of the cost function, when testing 1,2 and 3 iterations is 3. But 3 is still much smaller to the typical number of iterations usually used for such minimizations, and the experiments and analysis of this paper do not show that the minimization has converged after 3 iterations (Figure 9b even imply the opposite)."

Response

Sensitivity experiments (the following figure Fig. 2) told us that we really need 7~8

iterations to completely reach the minimization of the cost function. Meanwhile, we also found that the results of three iterations are not much different from the results of more iterations. As such, limited by computational cost, we choose Imax=1, 2 and 3 as the maximum NLS-4DVar iteration number in this sensitivity experiments and Imax=3 performs best as given on last paragraph of 3.3.3 Sensitivity experiments. The opposite performance in Fig. 9b of our manuscript mainly because of spin-up process of $CO_2$ and flux as describe in second paragraph of 3.3.2 Flux.

"It is important to note that the authors forget to say that their system is a global inversion system (and even to say that it inverts natural $CO_2$ fluxes, until the details of the equations clarifies it) which should strongly influence the way the problem of the initial condition should be tackled, the choice of the inversion cycles and of the data assimilation windows, the size of the ensembles and the number of iterations used for the inversions... in a more general way, the authors ignore the influence of the specific framework of their inversions -domain, resolution, data assimilated- on their results and on their choices of values for the inversion parameters."

Response

We added it in the first sentence of abstract and the title, see page 1 line 12: "Here we introduce a new version of the global carbon cycle data assimilation system". Tan-Tracker (v1) was based on GEOS-Chem, so it has same domain and resolution of it, as described in 3.1 Model settings and observations, and observation information can also be found there. Considering the selection of parameters, we show two main sensitivity experiments only to evaluate flux assimilation pass window length and maximum NLS-4DVar iteration number shown in fig.8 and fig.9 of our manuscript. The sensitivity experiments of the flux assimilation pass localization radius were used to select a localization radius of 1000 km (denoted as Loc-1k), 2000 km (denoted as Loc-2k), or 4000 km (denoted as Loc-4k); the other TT_v1 parameters remained unchanged. The flux and concentration results are shown in following figure Fig. 3. The time series of the monthly total flux (Fig. 3a) and the $CO_2$ concentration (Fig. 3c) results showed

that the assimilation results is better with 2000 km localization radius. It is reasonable to be longer than 900km in Carbon-Tracker (Peters et al., 2005) and Tan-Tracker (v0) because of a shorter model integration time meaning lower error from remote location. To be clear, we add this experiments on page 14 line 15 and Fig. 10 on page 31. The selection of localization parameters is based on the limit of computational cost (Zhang and Tian 2018).

"I add, without entering into too much details about it, that the quality of the text is not sufficient for a scientific publication. The abstract already gives a good illustration of the confusing way with which this manuscript is written. The authors insert a lot of technical jargon from the data assimilation community, but they actually misuse many of the corresponding terms (few examples: "surface flux inversion measurement", "flux assimilation" to speak about flux inversion assimilating CO2 data, the alternative use of "background" or "prior" to speak about the same thing, "one-step iteration", "atmospheric chemical transmission mode", "ensemble-based hybrid assimilation algorithm"...), leading to meaningless of confusing sentences. The abusive use of words and mathematical notations that seem more complicated than needed (or that they just forget to define, such as Py in eq 15) severely hampers the clarity of the paper."

Response

a. "surface flux inversion measurement" on page 2 line 14, 20; page 3 line 8 , 14 and Page 4 line 12 was changed to "surface flux inversion". And "CO2 concentration measurement" on page 15 line 10 was changed to "CO2 concentration".

b. Change "flux assimilation" to "flux assimilation pass" in our manuscript.

c. "Background flux" is different to "prior flux" in our manuscript as shown in Eq. 1. "background flux" served as the assimilation background field and where "prior flux" means prior flux data sets. To be clear, we add an explanation on page 4 line 21: "Note that for one certain assimilation cycle, "background flux" is different to "prior flux" as shown in Eq. 1; "background flux" served as the assimilation background field where

"prior flux" means prior flux data sets." To be clear, "prior (background) flux" on page 11 line 5 was changed to "prior flux"; "background results" on page 11 line 21 was changed to "prior results"; "direct background simulations" on page 15 line 15 was changed to "prior simulations".

d. "one-step iteration" on page 1 line 24 and on page 2 line 27 were changed to "one iteration".

e. "atmospheric chemical transmission mode" on page 3, line 12 was changed to "atmospheric chemical transport model".

f. "ensemble-based hybrid assimilation algorithm" on page 4, line 15 was changed to "ensemble-based assimilation algorithm".

g. Description of Py can be find on page 7 line 2, and now moved below Eq. 15.

Sorry for the inconvenience. After looking through our manuscript we changed some other errors, Listed below:

h. On page 2, line 9: "The surface carbon flux inversion method, especially carbon cycle data assimilation, obtained by combining model and atmospheric CO2 information, has made great progress"

i. On page 2, line 13: "Many attempts have been made, assimilating atmospheric CO2 measurements, to optimize surface carbon flux"

j. On page 2, line 25: "The same window lengths make uncertainties of CO2 and surface flux indistinguishable."

k. On page 2, line 30: "Several unconventional data assimilation attempts have been explored."

l. On page 3, line 19: "Specifically, the first performed CO2 assimilation pass uses a shorter window of 3 days to reduce the influence of background flux on the initial CO2"

m. On page 4, line 5: "Based on the NLS-4DVar assimilation method, assimilating satellite column-average CO2 concentration measurements of XCO2"

n. On page 4, line 11: "As such, the evolution of the CO2 concentration in the assimilation window is dominated by the background flux to improve the accuracy of surface flux inversion."

o. On page 5, line 3: "Considering computational cost, we chose N = 36."

p. On page 12, line 24: "However, in OSSEs, we can quantitatively analyze the prior flux, optimized flux and real flux to give the most direct judgment."

"Most of the introduction sounds like a random sampling of references to past inversions, with meaningless comments (like "the surface carbon flux inversion method, obtained by combining model and atmospheric CO2 information, has made great progress in carbon cycle data assimilation", "For example, CarbonTracker is a well-designed carbon assimilation system", "Basu et al. showed that satellite data provided an effective constraint for surface carbon source-sink inversion"...). It hardly provides clues about the specific topic of this paper. The analyzes in section 3.3 lack of depth and of hindsight on the significance and scope of the OSSEs and of the conclusions."

Response

We try to present in the introduction the development of carbon cycle data assimilation based on different observations, from in situ to aircraft to satellite measurements. After that, we introduce the assimilation system Tan-Tracker (v1). We added a discussion section in our manuscript, please see 4. Discussion.

Reference

Moradkhani H, Hsu K L, Gupta H, et al. Uncertainty assessment of hydrologic model states and parameters: Sequential data assimilation using the particle filter[J]. Water resources research, 2005a, 41(5). Moradkhani H, Sorooshian S, Gupta H V, et al. Dual state–parameter estimation of hydrological models using ensemble Kalman filter[J].

Advances in water resources, 2005b, 28(2): 135-147. Vrugt J A, Diks C G H, Gupta H V, et al. Improved treatment of uncertainty in hydrologic modeling: Combining the strengths of global optimization and data assimilation[J]. Water resources research, 2005, 41(1). Xu T R, Liu S M, Xu Z W, et al. A dual-pass data assimilation scheme for estimating surface fluxes with FY3A-VIRR land surface temperature[J]. Science China Earth Sciences, 2015, 58(2): 211-230.

Please also note the supplement to this comment:
https://www.geosci-model-dev-discuss.net/gmd-2019-54/gmd-2019-54-AC2-supplement.pdf

[Figure]

**Fig. 1.** Sensitivity experiment results: a. monthly total flux; b. monthly total flux deviation; c. daily root-mean-square error (RMSE) of CO2 concentration between the simulation/assimilation results and True

[Figure]

**Fig. 2.** Iteration time series of cost function value. (TT_v1 assimilation experiments with 10 iterations on flux assimilation pass performed on four consecutive windows, w1 to w4, from June to AugustïijL'

[Figure]

**Fig. 3.** Localization radii Sensitivity experiment results. ïïjĹsame to Fig. 1ïïjĽ

---

## Referee Comment (RC2) · Anonymous Referee #2 · 20 Sep 2019

Referee Comments on Han & Tian, "A dual-pass carbon cycle data assimilation system to estimate surface CO2 fluxes and 3D atmospheric CO2 concentrations from spaceborne measurements of atmospheric CO2"

This paper presents a new version (v1) of the Tan-Tracker carbon data assimilation system. The previous version (v0), as detailed in Tian, et al. (2014), was a forward-running filter with a 5-7 week assimilation window similar to the CarbonTracker scheme (Peters et al. 2005), with both CO2 flux and CO2 concentrations being solved for simultaneously. This version splits this into a two-step process, with CO2 concentration being

estimated first across a 3-day window, then CO2 fluxes inferred from those across a 2-week window.

Since the chief problem with these forward-running assimilation schemes in the past has been the shortness of the assimilation window, which truncates the span across which the dynamical constraint between fluxes and concentrations provided by the atmospheric transport model can operate, it is not clear what benefit is gained by both shortening this assimilation window and removing the flux constraint in the first step of the assimilation. The explanation given in the text for this ("to reduce the influence of the background flux on the initial CO2 concentration") does not make sense to this reviewer: it would seem more logical to use a longer assimilation window, so that the background fluxes (those before the start of the assimilation window) play less of a role, being further back in time with respect to the measurements being assimilated in the window. Given these issues, I am perplexed that the OSSE results presented show that this newer approach is somehow giving results that are closer to the truth than before – I would have thought that a properly-designed OSSE would show the opposite.

Measurement information in this scheme, as in CarbonTracker, can only be propagated backwards to previous times as far as the length of the assimilation window (here, two weeks (or 3 days?), in v0 5-7 weeks, in CarbonTracker 5 weeks, though this has been increased to 3-6 months in the latest annual release). The measurement information can only modify fluxes within this window – any corrective information coming from earlier fluxes is then mis-attributed to fluxes inside this span. This attribution or localization error grows worse as the assimilation window is shortened. In the past when only in situ data (mostly at the surface) was available, this was a significant problem: in many areas of the world, CO2 fluxes would not be "seen" at the measurement sites until many weeks, or even months, had transpired (think fluxes from the tropical land regions, the effect of which would be transported up due to the mainly convective transport there and not be seen until it came down later at higher latitudes or else got lucky enough to

be observed by an airplane before then). In this case, the localization of these far-field fluxes to the near-field around the measurements would cause large flux errors. Now, with satellite data this is less of an issue (there is generally an overpass of the satellite within 400 km of any spot on the globe at least once a week), but clouds and high aerosol conditions in the tropics reduce coverage there and suggest an assimilation window of at least several weeks would still be wise; the ability of the satellites to see a signal anywhere in the column both helps see the influence of the surface fluxes but also hurts by making it more difficult to say where the signal came from. And systematic errors (biases) in the satellite data further limit its usefulness. Given this, I would think that the powerful dynamical constraint provided by the transport models should not be cast aside by using assimilation windows as short as is done here. (Yes, the transport models have their own inaccuracies, but one can always use several of them to get an idea of their influence.)

The estimation of the $CO_2$ fields and $CO_2$ fluxes in two steps also seems problematic to me. Estimating the $CO_2$ fields first without the flux constraint seems to allow one to throw out the mass balance imposed by the transport model for the previous fluxes completely. The error caused by this should them project into the second step in which the fluxes are estimated. The two-step process would seem to eliminate the ability to solve for correlations between errors in the fluxes and errors in the initial concentration field.

In my view, the direction taken here towards shorter assimilation windows and a looser constraint from the transport model seems to be misguided. I would be more interested in the OSSEs quantifying the truncation errors incurred from these short windows, rather than what is shown here (I can't understand why the OSSEs give better results and I am suspicious that the OSSE setup does not capture all the relevant errors). I would not be surprised if $CO_2$ flux results obtained with this new v1 TanSat system are similar to those given by the old "mass-balance" methods of 20+ years ago: noisy to the point of making it difficult to identify the actual flux signal beneath the noise.

---

## Author Comment (AC3) · 27 Sep 2019

Responses to Anonymous Referee #2

We would like to thank the reviewer for carefully reading our paper and for all the valuable questions/comments/suggestions. We have thoroughly studied the reports and made a substantial revision to our paper by incorporating all suggestions given in the reports. Below we give our itemized responses.

Comment:

[Figure]

"Since the chief problem with these forward-running assimilation schemes in the past has been the shortness of the assimilation window, which truncates the span across which the dynamical constraint between fluxes and concentrations provided by the atmospheric transport model can operate, it is not clear what benefit is gained by both shortening this assimilation window and removing the flux constraint in the first step of the assimilation. The explanation given in the text for this ("to reduce the influence of the background flux on the initial $CO_2$ concentration") does not make sense to this reviewer: it would seem more logical to use a longer assimilation window, so that the background fluxes (those before the start of the assimilation window) play less of a role, being further back in time with respect to the measurements being assimilated in the window. Given these issues, I am perplexed that the OSSE results presented show that this newer approach is somehow giving results that are closer to the truth than before – I would have thought that a properly-designed OSSE would show the opposite."

Response:

The window length of one assimilation system is systematically determined by the forecast model (the Atmospheric Chemistry Transport Model), model resolution, observation, assimilation algorithm and system framework. Therefore, the assimilation window length is usually not fixed in different systems.

Table 1 shows that the Tan-Tracker (v1) is substantially different from the Carbon Tracker (CT) and even its previous version (Tan-Tracker, v0). This is not surprising that they have different window lengths because of the above significant differences in the two assimilation systems. Carful comparisons also show that the Tan-Tracker (v0) is much similar to Carbon Tracker (CT), which leads to their similar window lengths.

However, despite that, we don't think the window length should be fixed in one single assimilation system. It will likely change as the incoming observations and the resolutions, which usually should be determined through sensitivity experiments (as done in

this work).

In addition, it is not a technically difficult task to lengthen the assimilation window in Tan-Tracker (v1). It's precisely because that our sensitivity experiments choose the two weeks as the final OSSEs.

Tan-Tracker (v1) uses dual-pass assimilation strategy to assimilate satellite XCO2 observations which can reduce the impact of localization error caused by sparse observation and accumulated error of historical optimized fluxes. In the current assimilation window, the simulated CO2 concentration errors originated from both the initial CO2 at the beginning of this window and the background flux of this window (see page 4 line 21). More specifically, the initial CO2 contains accumulated error of the historical optimal fluxes also contains error of the start initial CO2 (the first window). These errors entangled with the model evolution, which is indeed difficult to optimize the CO2 concentrations and fluxes altogether. In the dual-pass assimilation strategy, a shorter CO2 assimilation pass (3-day) is first adopted to optimize the initial CO2 concentrations (shorter window can reduce the influence of background flux of this window on initial CO2 assimilation). The optimized initial CO2 concentrations at the beginning of this window are then used for the following flux assimilation pass (14-day), which is a reasonable strategy to differentiate the errors from the initial CO2 and the background fluxes (For more details, please see abstract, section 2.1, section 4 and Fig.1).

Furthermore, to guarantee "mass-balance", we start the update section by the background initial CO2 (instead of the optimized initial CO2 concentrations obtained in the first assimilation pass) forced by optimized fluxes obtained in the second assimilation pass.

Comment:

"Measurement information in this scheme, as in CarbonTracker, can only be propagated backwards to previous times as far as the length of the assimilation window (here, two weeks (or 3 days?), in v0 5-7 weeks, in CarbonTracker 5 weeks, though this

has been increased to 3-6 months in the latest annual release). The measurement information can only modify fluxes within this window – any corrective information coming from earlier fluxes is then mis-attributed to fluxes inside this span. This attribution or localization error grows worse as the assimilation window is shortened. In the past when only in situ data (mostly at the surface) was available, this was a significant problem: in many areas of the world, CO2 fluxes would not be "seen" at the measurement sites until many weeks, or even months, had transpired (think fluxes from the tropical land regions, the effect of which would be transported up due to the mainly convective transport there and not be seen until it came down later at higher latitudes or else got lucky enough to be observed by an airplane before then). In this case, the localization of these far-field fluxes to the near-field around the measurements would cause large flux errors. Now, with satellite data this is less of an issue (there is generally an overpass of the satellite within 400 km of any spot on the globe at least once a week), but clouds and high aerosol conditions in the tropics reduce coverage there and suggest an assimilation window of at least several weeks would still be wise; the ability of the satellites to see a signal anywhere in the column both helps see the influence of the surface fluxes but also hurts by making it more difficult to say where the signal came from. And systematic errors (biases) in the satellite data further limit its usefulness. Given this, I would think that the powerful dynamical constraint provided by the transport models should not be cast aside by using assimilation windows as short as is done here. (Yes, the transport models have their own inaccuracies, but one can always use several of them to get an idea of their influence.)"

Response:

The observation coverage, column averaged CO2 observations and systematic errors are problems encountered when using satellite data. Tan-Tracker (v1) mitigates the side effects by using proper window length and observation operator (Please see the above response and section 2.1, section 4 and Fig.1).

OCO-2 can provide a large amount of XCO2 observations with high spatial and temporal resolution. There are about 10ˆ5 observations per day, meaning 10ˆ6~10ˆ7 observations per flux assimilation window (14-day). After rigorous data quality control and data thinning, about 10ˆ5~10ˆ6 observations will be assimilated into Tan-Tracker (v1) within one flux assimilation pass (see Fig.2). And the sun-synchronous orbit of OCO-2 has a 16-day (233 orbit) ground track repeat cycle. This means that the appropriate 14-day flux assimilation window has a better global observations coverage with less repeat location (see Fig.2).

Moreover, column averaged observations XCO2 can influence each model level by using observation operator (For more details, please see section 3.1 and Eq.31). In current Tan-Tracker (v1), we mainly focus on assimilating satellite data and adopting proper strategy to find appropriate surface flux inversion; In the future, we will use multi-source CO2 observations (including in-situ observations) and be more accurate in resolving surface flux signal.

Additionally, we did not consider the observation systematic errors in OSSEs (For more details, please see section 3.1), but we have used OCO-2 data with bias correction in real assimilation experiments; And we believe that the XCO2 retrieval team will make continuous efforts to eliminate the systematic bias (OCO-2 Science Team, 2017; O'Dell et al., 2018).

Using the novel dual-pass assimilation strategy and assimilating OCO-2 satellite observations, Tan-Tracker (v1) does not require a long assimilation window. This conclusion can also be drawn from OSSEs of window length (For more details, please see section 3.3.3).

Comment:

"The estimation of the CO2 fields and CO2 fluxes in two steps also seems problematic to me. Estimating the CO2 fields first without the flux constraint seems to allow one to throw out the mass balance imposed by the transport model for the previous fluxes completely. The error caused by this should them project into the second step in which

the fluxes are estimated. The two-step process would seem to eliminate the ability to solve for correlations between errors in the fluxes and errors in the initial concentration field.

In my view, the direction taken here towards shorter assimilation windows and a looser constraint from the transport model seems to be misguided. I would be more interested in the OSSEs quantifying the truncation errors incurred from these short windows, rather than what is shown here (I can't understand why the OSSEs give better results and I am suspicious that the OSSE setup does not capture all the relevant errors). I would not be surprised if CO2 flux results obtained with this new v1 TanSat system are similar to those given by the old "mass-balance" methods of 20+ years ago: noisy to the point of making it difficult to identify the actual flux signal beneath the noise."

Response:

The "mass balance" issue of Tan-Tracker (v1) is not clearly explained in the previous version of this manuscript. To state it clear, we modify a confusing use of Ca to Cb in Fig.1, add "background initial CO2" in Page 5 line 7 and add the descriptions in Page 15 line 7.

Mass balance is important for a carbon cycle assimilation system. In Tan-Tracker (v1), CO2 assimilation pass is a directly change to the atmospheric carbon pool and will result in flux bias if accumulated through the whole assimilation process. To avoid this, the update section starts from the background initial CO2; As a result, the analysis CO2 concentrations are forced by the model and optimized fluxes only, starting from background initial CO2 of first window. This also means chemical transport model can impose continuous constraint on flux and CO2 without truncation error. In other words, optimized flux is not only the best-fitting of current window constrained by observations under low initial CO2 error, but the best-fitting of the whole assimilation progress constrained by model and mass balance.

reference

O'Dell, C. W., Eldering, A., Wennberg, P. O., Crisp, D., Gunson, M. R., Fisher, B., Frankenberg, C., Kiel, M., Lindqvist, H., Mandrake, L., Merrelli, A., Natraj, V., Nelson, R. R., Osterman, G. B., Payne, V. H., Taylor, T. E., Wunch, D., Drouin, B. J., Oyafuso, F., Chang, A., McDuffie, J., Smyth, M., Baker, D. F., Basu, S., Chevallier, F., Crowell, S. M.R., Feng, L., Palmer, P. I., Dubey, M., Garcia, O. E., Griffith, D. W.T., Hase, F., Iraci, L. T., Kivi, R., Morino, I., Notholt, J., Ohyama, H., Petri, C., Roehl, C. M., Sha, M. K., Strong, K., Sussmann, R., Te, Y., Uchino, O. & Velazco, V. A. (2018). Improved retrievals of carbon dioxide from Orbiting Carbon Observatory-2 with the version 8 ACOS algorithm. Atmospheric Measurement Techniques, 11 (12), 6539-6576.

Please also note the supplement to this comment:
https://www.geosci-model-dev-discuss.net/gmd-2019-54/gmd-2019-54-AC3-supplement.pdf
* * *
**Table 1. Differences between Tan-Tracker (v1), Tan-Tracker (v0) and CT**

| | Tan-Tracker (v1) | Tan-Tracker (v0) | Carbon Tracker (CT) |
|---|---|---|---|
| **Forecast model** | GEOS-Chem | GEOS-Chem | TM5 |
| **Assimilation algorithm** | (the nonlinear least squares) 4DVar (take the forecast model as the strong constraint), to assimilate all the observations simultaneously | POD-4DVar (actually one Kalman smoother, see Tian et al., 2018) | EnSRF (Very similar to Kalman smoother), to assimilate observations one by one |
| **Observations** | Satellite XCO2 (about $10^5$ observations per day) | In-situ (about $10^3$ per week) | In-situ (about $10^3$ per week) |
| **Assimilation framework** | Dual-pass assimilation framework (3days, 2 weeks) | Joint-data assimilation framework | Observation window + lag window |
| **Localization scheme** | Efficient localization scheme proposed by Tian et al. (2018) and Zhang and Tian (2018) | A simple localization scheme (Tian and Feng, 2015) | Through: $\mathbf{P} = \mathbf{P} \cdot \mathbf{L}$ $\mathbf{L} = e^{-d_{ij}/l}$ |

**Fig. 1.**

**Supplement:**

[revised manuscript text omitted]
_2$. Note that for one certain assimilation cycle, "background flux" is different to "prior flux" as shown in Eq. 1; "background flux" served as the assimilation background field where "prior flux" means prior flux data sets. $H_k$ is a satellite XCO2 observation operator, as given in Eq. (31). Putting $\mathbf{U}_s, \mathbf{U}_b, H_k$ together with observations $X_{CO_2,Obs}$ into the NLS-4DVar processor, we can obtain an optimized initial $CO_2$ $\mathbf{U}_{a,t_0}$,

25   to be used as the initial $CO_2$ of the flux assimilation pass.

In the flux assimilation pass (the red portion shown in Fig. 1), we assume that there is no error in anthropogenic emissions, and only optimize the terrestrial ecosystems flux and oceans flux:

$$\mathbf{F}^* = \mathbf{F}^*_{bio} + \mathbf{F}^*_{oce}, \tag{2}$$

where $\mathbf{F}^*$ is the prior flux, with *bio* referring to the flux from the terrestrial biosphere, and *oce* representing the flux from the

30   ocean. Starting from the optimized initial $CO_2$ $\mathbf{U}_{a,t_0}$, forcing by a set of prepared flux ensembles:

$$F_{s,i} = \lambda_{s,i} \times F^*, (i = 1, \cdots, N),$$
(3)

we obtain a set of 2-week $CO_2$ ensembles $U_{s,i}, (i = 1, \cdots, N)$, where $\lambda_{s,i} (i = 1, \cdots, N)$ is a set of scale factors (see Section 2.3). Considering computational cost, we chose $N = 36$. Simultaneously, starting from the background initial $CO_2$ $U_{b,t_0}$ forcing by the background flux $F_b = \lambda_b \times F^*$, we simulated the 2-week $CO_2$

5 concentration $U_b$ as background $CO_2$. Putting $\lambda_s$, $U_s$, $\lambda_b$, $U_b$, $H_k$, and observations $X_{CO_2,Obs}$ into the NLS-4DVar processor, we can obtain the optimized scale factor $\lambda_a$, with the optimized flux given by $F_a = \lambda_a \times F^*$.

The update section is shown as the black portion of Figure 1. Starting from the  background initial $CO_2$ $U_{b,t_0,r}$ (To guarantee the system's mass-balance) of the $r$th assimilation cycle forced by optimized fluxes $F_{a,r}$, and integrating through the window of the flux assimilation pass to the end, we obtain the background initial $CO_2$ concentration $U_{b,t_0,r+1}$ of the $(r+1)$th

10 assimilation cycle. Unlike the joint Tan-Tracker (v0) system, the background initial $CO_2$ concentration of Tan-Tracker (v1) is obtained by a running model, as opposed to a direct assimilation, thus eliminating the problem of $CO_2$ over-optimization. Similar to the approach of Peters (2007), the $(r+1)$th background flux, $F_{b,r+1} = \lambda_{b,r+1} \times F^*_{r+1}$, is applied using the mean value of the two previous time steps' scale factors $a$:

$$\lambda_{b,r+1} = (\lambda_{a,r} + \lambda_{a,r-1} + 1)/3.$$
(4)

15 **2.2 Coupling of NLS-4DVar with Tan-Tracker (v1) assimilation framework**

The NLS-4DVar algorithm is used to solve the optimal initial perturbation $x'_a$ to satisfy the incremental form of the 4DVar cost function:

$$J(x') = \frac{1}{2}(x')^T B^{-1}(x') + \frac{1}{2}\sum_{k=0}^{S}\left[L'_k(x') - y'_{obs,k}\right]^T R_k^{-1}\left[L'_k(x') - y'_{obs,k}\right],$$
(5)

where $x' = x - x_b$ is the perturbation of the background field $x_b$ at initial time $t_0$, and

20 $$L'_k(x') = L_k(x_b + x') - L_k(x_b),$$
(6)

$$y'_{obs,k} = y_{obs,k} - L_k(x_b),$$
(7)

[revised manuscript text omitted]
. Note that, 14-days flux assimilation pass window length is close to those adopted by some other published inversion systems, such as the one week length of Carbon-Tracker (Peters et al., 2007), the one-month length of Basu et al. (2013) also the 7-days length of Tan-Tracker (v0) (Tian et al., 2014). The window of the appropriate length (14 days) had a small initial $CO_2$ concentration deviation, the appropriate integration time, and was closest to the OCO-2 satellite 16-day regression period, i.e., it was possible to absorb more observations to obtain good flux inversion results.

As the maximum NLS-4DVar iteration number increases, the assimilation results tend to converge, especially for solving the problem of high nonlinear systems. However, the computational cost increases with the number of iterations. The sensitivity experiments of the maximum NLS-4DVar iteration number selected one (*Imax = 1*), two (*Imax = 2*), and three (*Imax = 3*) iterations, with the remaining parameters retaining the values of *TT_v1*. The resulting flux and concentration results are shown in Figure 9. The time series of the monthly total flux (Fig. 9a) and the $CO_2$ concentration (Fig. 9c) results showed that the assimilation results improved and tended to converge quickly as the number of maximum NLS-4DVar iterations increased. Considering the computational cost, we chose three maximum NLS-4DVar iterations as the final solution.

The sensitivity experiments of the flux assimilation pass localization radii were used to select a localization radius of 1000 km (denoted as Loc-1k), 2000 km (denoted as Loc-2k), or 4000 km (denoted as Loc-4k); the other TT_v1 parameters remained unchanged. The flux and concentration results are shown in Fig. 10. The time series of the monthly total flux (Fig. 10a) and the $CO_2$ concentration (Fig. 10c) results showed that the assimilation results is better with 2000 km localization radii. It is reasonable to be longer than 900km in Carbon-Tracker (Peters et al., 2005) and Tan-Tracker (v0) because of a shorter model integration time meaning lower error from remote location.

**4. Discussion**

For each assimilation cycle, the simulated $CO_2$ concentration errors originated from both the initial $CO_2$ and the background flux errors. These errors entangled with the model evolution, which is indeed difficult to optimize the $CO_2$ concentrations and fluxes altogether. Dual-pass assimilation system Tan-Tracker (v1) was proposed to proper distinguish the errors and reduce their influences. The $CO_2$ assimilation pass with a shorter length (3-days) window is firstly utilized to assimilate the initial $CO_2$ concentrations with little influence of the background fluxes. This allows us to initiate the subsequent flux assimilation

pass from the optimal initial $CO_2$ concentration. A properly elongated 2-weeks length is specially designed to incorporate enough observations for surface fluxes. Through the above optimization steps, initial $CO_2$ errors and background flux errors are properly distinguished; $CO_2$ concentration and surface $CO_2$ flux are mutually adjusted and optimized. In the ensemble-based Tan-Tracker system, the uncertainties are described by the ensembles in the NLS-4DVar, which are further optimized with the ensembles update. In a summary, dual-pass presents a proper way controlling both $CO_2$ initial condition and flux successively.

Mass balance is important for a carbon cycle assimilation system. In Tan-Tracker (v1), $CO_2$ assimilation pass is a directly change to the atmospheric carbon pool and will result in flux bias if accumulated through the whole assimilation process. To avoid this, the update section starts from the background initial $CO_2$; As a result, the analysis $CO_2$ concentrations are forced by the model and optimized fluxes only, starting from background initial $CO_2$ of first window. This also means chemical transport model can impose continuous constraint on flux and $CO_2$ without truncation error. In other words, optimized flux is not only the best-fitting of current window constrained by observations under low initial $CO_2$ error, but the best-fitting of the whole assimilation progress constrained by model and mass balance.

[revised manuscript text omitted]
 English in this document has been checked by at least two professional editors, both native speakers of English. For a certificate, please see:

http://www.textcheck.com/certificate/0ZjjbP